# Context-free Recognition with Transformers

**Selim Jerad** [1]   **Anej Svete** [1]   **Sophie Hao** [2]   **Ryan Cotterell** [1]   **William Merrill** [3]

## Abstract

Transformers excel empirically on tasks that process well-formed inputs according to some grammar, such as natural language and code. However, it remains unclear how they can process grammatical syntax. In fact, under standard complexity conjectures, standard transformers cannot recognize context-free languages (CFLs), a canonical formalism to describe syntax, or even regular languages, a subclass of CFLs. Past work has shown that $\mathcal{O}(\log(N))$ *looping layers* (w.r.t. input length $N$) allow transformers to recognize regular languages, but the question of context-free recognition with looped transformers remained open. In this work, we show that looped transformers with $\mathcal{O}(\log(N))$ looping layers and $\mathcal{O}(N^6)$ padding symbols can recognize all CFLs. However, training and inference with $\mathcal{O}(N^6)$ padding symbols is potentially impractical. Fortunately, we show that, for natural subclasses such as unambiguous CFLs, the recognition problem on transformers becomes more tractable, requiring $\mathcal{O}(N^3)$ padding. Empirically, looped and padded transformers perform better than fixed-depth transformers in recognizing CFLs. Overall, our results shed light on the intricacy of CFL recognition by transformers: while general recognition may require an intractable amount of padding, natural constraints such as unambiguity yield efficient recognition algorithms.

## 1. Introduction

Transformers are proficient at many natural language ([Qin et al.](), [2024]) and coding ([Jiang et al.](), [2025]) tasks, both of which involve processing hierarchical structures. In formal language theory, this hierarchical structure is captured by context-free languages (CFLs), a canonical formal model

for the kind of syntax that natural language and code rely on. Empirical work supports this view: analysis of internal representations—syntactic probing—has shown that transformers learn to encode syntactic features relevant for *parsing*, the task of extracting the syntactic structure of a sentence ([Hewitt & Manning](), [2019]; [Arps et al.](), [2022]; [Zhao et al.](), [2023]). However, the literature lacks a precise theory of how transformers can implement CFL recognition. Our paper fills this gap.

More precisely, the problem we study is the following: Given a CFG $\mathcal{G}$, can a string $\boldsymbol{w}$ be generated by $\mathcal{G}$? Several foundational *serial* parsing algorithms ([Earley](), [1970]; [Cocke](), [1969]; [Kasami](), [1965]; [Younger](), [1967]) solve this problem efficiently—in $\mathcal{O}(|\boldsymbol{w}|^3)$ time. However, *parallel* parsing algorithms can run even faster given sufficient processors ([Ruzzo](), [1980]; [Rytter](), [1987]; [Lange et al.](), [1992]). Indeed, one might suspect *prima facie* that transformers' highly parallel, fixed-depth structure ([Merrill & Sabharwal](), [2023b]) enables them to efficiently parse strings. We therefore focus on implementing parallel parsing algorithms in transformers. To that end, we first note that even regular languages, a strict subset of CFLs, cannot be recognized by fixed-depth transformers under the standard complexity conjecture $\mathrm{TC}^0 \subsetneq \mathrm{NC}^1$.[1] *Looping* layers help: $\log(N)$ looping layers (where $N$ is the input length) allow transformers to recognize regular languages ([Merrill & Sabharwal](), [2024a]). Thus, our analysis will require looping to some extent. However, the question of whether logarithmic looping is sufficient for CFL recognition is still open. In this work, we answer it in the affirmative, showing that looping—along with additional *padding* symbols ([Merrill & Sabharwal](), [2025])—is sufficient to implement certain parallel CFG parsing algorithms.

Our first result shows via a direct construction that general CFL recognition can be expressed by looping layers $\mathcal{O}(\log(N))$ times and with $\mathcal{O}(N^6)$ padding symbols. We then ask whether simpler classes of CFLs can be recognized by transformers with fewer resources. Again, we find that the answer is affirmative. In particular, we identify *unambiguity* and *linearity* as two natural properties that make CFL recognition easier. Unambiguous CFLs, characterized by

---

[1]ETH Zürich [2]Boston University [3]Allen Institute for AI. Correspondence to: Selim Jerad <selimjerad@gmail.com>, William Merrill <willm@allenai.org>.

*Proceedings of the $43^{rd}$ International Conference on Machine Learning*, Seoul, South Korea. PMLR 306, 2026. Copyright 2026 by the author(s).

---

[1]Regular language recognition is complete for $\mathrm{NC}^1$ ([Barrington & Thérien](), [1988]) while fixed-depth transformers fall in $\mathrm{TC}^0$ ([Merrill et al.](), [2022]; [Chiang](), [2025]).

strings having at most one possible parse, allow for recognition with reduced padding but more looping. This aligns with transformers' struggles to parse ambiguous grammars in practice (Khalighinejad et al., 2023). Furthermore, imposing linearity (where each grammar rule has at most one non-terminal on its right-hand side) reduces the amount of looping and padding required for recognizing unambiguous CFLs. Our main results are summarized in Tab. 1.

We also empirically verify that looped and padded transformers can better process CFLs than fixed-depth ones. We evaluate fixed-depth and dynamically scaling transformers as recognizers of formal languages (Butoi et al., 2025), and find that the latter often outperforms the former on CFLs of various complexity (Tab. 2); the largest gains (from 4 to 8 points of accuracy) occur on harder CFLs such as $\mathrm{Dyck}(2)$, Palindrome and BFVP.

## 2. Preliminaries

An **alphabet** $\Sigma$ is a finite, non-empty set of **symbols**. A **string** $\boldsymbol{w} = w_1 \cdots w_N$ with $w_n \in \Sigma$ is a finite sequence of symbols. We write $|\boldsymbol{w}| = |w_1 \cdots w_N| = N$ for the length of $\boldsymbol{w}$. For integers $i, j$ with $i \leq j$, we use the standard interval notation $[i,j] \stackrel{\text{def}}{=} \{i, i+1, \ldots, j\}$, $(i,j] \stackrel{\text{def}}{=} \{i+1, \ldots, j\}$, $[i,j) \stackrel{\text{def}}{=} \{i, \ldots, j-1\}$, and $(i,j) \stackrel{\text{def}}{=} \{i+1, \ldots, j-1\}$. For any interval $I \subseteq \{1, \ldots, N\}$, we write $\boldsymbol{w}_I$ for the substring of $\boldsymbol{w}$ at positions in $I$. Explicitly, the four shapes expand to $\boldsymbol{w}_{[i,j]} \stackrel{\text{def}}{=} w_i w_{i+1} \cdots w_j$, $\boldsymbol{w}_{(i,j]} \stackrel{\text{def}}{=} w_{i+1} w_{i+2} \cdots w_j$, $\boldsymbol{w}_{[i,j)} \stackrel{\text{def}}{=} w_i w_{i+1} \cdots w_{j-1}$, and $\boldsymbol{w}_{(i,j)} \stackrel{\text{def}}{=} w_{i+1} w_{i+2} \cdots w_{j-1}$. Half-open intervals are convenient as we get $\boldsymbol{w}_{(i,j]} \, \boldsymbol{w}_{(j,k]} = \boldsymbol{w}_{(i,k]}$. The empty string, denoted $\varepsilon$, is the unique string of length 0. The set of all strings over an alphabet $\Sigma$—including the empty string—is given by its **closure**, denoted $\Sigma^*$. A **formal language** $L$ over $\Sigma$ is a subset of $\Sigma^*$. A **language recognizer** is a function $f$ with the signature $\Sigma^* \to \{0, 1\}$. $f$ **accepts** a string $\boldsymbol{w}$ iff $f(\boldsymbol{w}) = 1$. $f$ **recognizes** a language $L \subseteq \Sigma^*$ when $f(\boldsymbol{w}) = 1$ iff $\boldsymbol{w} \in L$.

### 2.1. Context-free Grammars

**Definition 2.1.** *A **context-free grammar** (CFG) $\mathcal{G}$ is a tuple $(\mathcal{N}, \Sigma, \mathcal{P}, \mathrm{S})$ where:*

- *$\mathcal{N}$ is a finite, nonempty set of **nonterminals**;*
- *$\Sigma$ is a finite alphabet of **terminals**, disjoint from $\mathcal{N}$;*
- *$\mathcal{P} \subseteq \mathcal{N} \times (\mathcal{N} \cup \Sigma)^*$ is a finite set of **productions** of the form $\mathrm{A} \to \boldsymbol{\alpha}$, where $\mathrm{A} \in \mathcal{N}$ and $\boldsymbol{\alpha} \in (\mathcal{N} \cup \Sigma)^*$;*
- *$\mathrm{S} \in \mathcal{N}$ is a distinguished **start symbol**.*

*We denote terminal and nonterminal symbols by lowercase and uppercase symbols, respectively.*

A string of non-terminals and terminals $\boldsymbol{\alpha} \in (\mathcal{N} \cup \Sigma)^*$ is

a **sentential form**. A CFG generates strings by repeatedly applying rules to sentential forms derived from the start symbol until it produces a sequence of terminal symbols, i.e., a **string**. For sentential forms $\boldsymbol{\alpha}, \boldsymbol{\beta} \in (\mathcal{N} \cup \Sigma)^*$, we write $\boldsymbol{\alpha} \Rightarrow \boldsymbol{\beta}$ if there exist sentential forms $\boldsymbol{\alpha}_1, \boldsymbol{\alpha}_2$, a non-terminal $\mathrm{A} \in \mathcal{N}$, and a rule $\mathrm{A} \to \boldsymbol{\gamma} \in \mathcal{P}$ such that $\boldsymbol{\alpha} = \boldsymbol{\alpha}_1 \, \mathrm{A} \, \boldsymbol{\alpha}_2$ and $\boldsymbol{\beta} = \boldsymbol{\alpha}_1 \, \boldsymbol{\gamma} \, \boldsymbol{\alpha}_2$. The derivation relation $\stackrel{*}{\Rightarrow}$ is the reflexive, transitive closure of $\Rightarrow$. We say a non-terminal $\mathrm{A}$ **derives** a string $\boldsymbol{w} \in \Sigma^*$ if $\mathrm{A} \stackrel{*}{\Rightarrow} \boldsymbol{w}$, i.e. $\boldsymbol{w}$ is the derivation's **yield**. The **language of a CFG** $\mathcal{G}$ is the set $L(\mathcal{G}) \stackrel{\text{def}}{=} \{\boldsymbol{w} \in \Sigma^* \mid \mathrm{S} \stackrel{*}{\Rightarrow} \boldsymbol{w}\}$. A formal language is **context-free** if it is the language of some $\mathcal{G}$.

It is common practice to consider CFGs in a normal form. A CFG $\mathcal{G}$ is in **Chomsky normal form (CNF)** if any $p \in \mathcal{P}$ is either of the form $\mathrm{A} \to \mathrm{BC}$, $\mathrm{A} \to a$ or $\mathrm{S} \to \varepsilon$. Every CFL can be described by a CFG in CNF.

### 2.2. Transformers

We consider Merrill & Sabharwal's (2025) idealization of the transformer architecture (Merrill & Sabharwal, 2024a).

**Definition 2.2** (Averaging hard-attention transformer). *An **averaging hard-attention transformer** (AHAT) $\mathsf{T}$ of width $D$ and depth $L$ is a tuple $\mathsf{T} = (\mathsf{emb}, (\mathcal{L}^{(\ell)})_{\ell=1}^L, c)$ where:*

- *$\mathsf{emb} \colon \Sigma^* \to (\mathbb{Q}^D)^*$ is a position-wise **input embedding** that maps each input symbol to a vector in $\mathbb{Q}^D$;*
- *each **layer** $\mathcal{L}^{(\ell)} \colon (\mathbb{Q}^D)^* \to (\mathbb{Q}^D)^*$ is a length-preserving map that composes a layer-normalization, an attention sublayer whose attention weights are computed by the averaging hard-attention projection* hardmax, *and a position-wise feedforward network;*
- *$c \colon \mathbb{Q}^D \to \{0, 1\}$ is a **classification function**.*

*The transformer first computes the length-preserving mapping $\mathcal{L}^{(L)} \circ \cdots \circ \mathcal{L}^{(1)} \circ \mathsf{emb}(\boldsymbol{w})$. Let $\boldsymbol{x}_{\mathrm{EOS}}^L \in \mathbb{Q}^D$ be the final contextual representation of* EOS *after the $L$ transformer layers. We say the transformer accepts $\boldsymbol{w}$ iff $\mathsf{T}(\boldsymbol{w}) \stackrel{\text{def}}{=} c(\boldsymbol{x}_{\mathrm{EOS}}^L) = 1$.*

We refer to §A for the formal definitions of each component of the transformer.

Average hard-attention returns a uniform average of the values of symbols that maximize the attention score.[2] The transformers use *multi*-pre-norm, where the layer normalization is applied before the residual connection on either the entire hidden state or on distinct subsets thereof (Merrill & Sabharwal, 2024b). We assume logarithmic-precision arithmetic, where computations are performed with $\mathcal{O}(\log(N))$ bits for an input of size $N$. Coupling AHATs and log-precision un-

---

[2]In practice, the training dynamics of standard transformers may push some attention heads toward approximating average-hard attention (Merrill et al., 2021), though this behavior is not universal.

| Language class | Padding symbols required | Looping layers required | Reference |
|---|---|---|---|
| General CFLs | $\mathcal{O}(N^6)$ | $\mathcal{O}(\log(N))$ | Thm. 3.1 |
| Unambiguous CFLs | $\mathcal{O}(N^3)$ | $\mathcal{O}(\log^2(N))$ | Thm. 4.1 |
| Unambiguous linear CFLs | $\mathcal{O}(N^2)$ | $\mathcal{O}(\log(N))$ | Thm. 4.2 |

*Table 1.* The computational resources required by transformers to recognize different classes of context-free languages (CFLs).

locks useful gadgets such as storing string indices, counting symbol occurrences across the string and performing equality checks across positions (Merrill & Sabharwal, 2024b). We assume that the input string is augmented with both the beginning-of-sequence (BOS) and end-of-sequence (EOS) symbols.

As defined in Def. 2.2, transformers have a *fixed* size, while general CFL recognition requires *dynamically-scaling* resources on classical models of parallel computation (Venkateswaran, 1991). We therefore also consider an extension of the transformer architecture where its size can *dynamically* increase. Concretely, we can first dynamically scale the number of layers[3] (Merrill & Sabharwal, 2024a).

**Definition 2.3.** *A $d(N)$-**looped** transformer $T$ is a tuple $T = (\mathrm{emb}, (\mathcal{L}^{(\ell)})_{\ell=1}^{L}, c, \ell_1, \ell_2)$ where:*

- *$(\mathrm{emb}, (\mathcal{L}^{(\ell)})_{\ell=1}^{L}, c)$ is an AHAT as defined in Def. 2.2;*
- *$\ell_1, \ell_2 \in (1, L)$ define a partition of the layers into $A \overset{\mathrm{def}}{=} (\mathcal{L}^{(\ell)})_{\ell=1}^{\ell_1-1}$, $B \overset{\mathrm{def}}{=} (\mathcal{L}^{(\ell)})_{\ell=\ell_1}^{\ell_2-1}$ and $C \overset{\mathrm{def}}{=} (\mathcal{L}^{(\ell)})_{\ell=\ell_2}^{L}$;*
- *Upon a forward pass, $B$ is repeated $\mathcal{O}(d(N))$ times for an input string of length $N$.*

*The transformer now computes the length-preserving mapping $C \circ (B)^{\mathcal{O}(d(N))} \circ A \circ \mathrm{emb}(w)$, and accepts $w$ via $c$.*

The amount of computation performed by self attention is definitionally quadratic in the string length. One can dynamically increase this by adding *padding space* (Merrill & Sabharwal, 2025).

**Definition 2.4.** *A (looped) transformer is $w(N)$-**padded** if $\mathcal{O}(w(N))$ padding symbols are appended to the end of the input string $w$, i.e. the input string is $w \underbrace{\square \cdots \square}_{\mathcal{O}(w(N))}$, where $\square$ is a distinguished padding symbol.*

Scaling number of layers and padding symbols in transformers is analogous to scaling time and space Boolean circuits (Merrill & Sabharwal, 2025), a classical parallel model of computation. Allowing for different looping and padding budgets results in different classes of transformers. We adopt naming conventions of these models from Merrill & Sabharwal (2025). We denote by $\mathrm{AHAT}_k^d$ the class of

---

[3]To guarantee the transformer width is constant while the number of layers grows with input length, we recall transformer layers can reset intermediate values in looping layers (Merrill & Sabharwal, 2024a).

languages recognized by $\log^d(N)$-looped, $N^k$-padded averaging hard-attention transformers with strict causal masking. We similarly denote by UAHAT average hard-attention transformers with no masking and MAHAT transformers that use both masked and unmasked attention heads. Conveniently, AHATs can simulate MAHATs:

**Lemma 2.1** (Merrill & Sabharwal, 2025, Prop. 1)**.** *For $d \geq 1$, $\mathrm{UAHAT}_k^d \subseteq \mathrm{MAHAT}_k^d \subseteq \mathrm{AHAT}_{1+\max(k,1)}^d$.*

## 3. Parallel Recognition of CFLs

We now present an algorithm for parallel CFL recognition, which synthesizes ideas from previous work on algorithms for parallel CFL recognition (Ruzzo, 1980; Rossmanith & Rytter, 1992; Lange & Rossmanith, 1990). We show in §3.3 how to implement this algorithm on AHATs.

### 3.1. Items and Parse Trees

Many parsing algorithms—including those we consider here—manipulate **items**, defined as follows. Given a grammar $\mathcal{G} = (\mathcal{N}, \Sigma, \mathcal{P}, \mathrm{S})$ and a string $w \in \Sigma^*$ of length $N$, an **item** is a tuple of one of four shapes:

$$(i, \mathrm{A}, j], \quad [i, \mathrm{A}, j], \quad [i, \mathrm{A}, j), \quad \text{or} \quad (i, \mathrm{A}, j),$$

with $\mathrm{A} \in \mathcal{N}$ and indices $i, j \in \{0, 1, \ldots, N\}$. An item is **realizable** if its non-terminal derives the substring determined by the bracket shape; e.g., $(i, \mathrm{A}, j] = \top \iff \mathrm{A} \overset{*}{\Rightarrow} w_{(i,j]}$, and analogously for $[i, \mathrm{A}, j]$, $[i, \mathrm{A}, j)$, $(i, \mathrm{A}, j)$. We write $\mathcal{I}$ for the set of all items associated with string $w$ and a grammar $\mathcal{G}$, i.e., all tuples of the four shapes above with non-terminals from $\mathcal{N}$ and indices in $\{0, 1, \ldots, N\}$. We suppress dependence on $w$ and $\mathcal{G}$ for notational simplicity.

Realizable items can be combined according to the rules of $\mathcal{G}$ to produce new items. In particular, if $\mathrm{A} \to \mathrm{BC} \in \mathcal{P}$, then for any indices $i < k < j$,

$$\frac{(i, \mathrm{B}, k] \quad (k, \mathrm{C}, j]}{(i, \mathrm{A}, j]},$$

i.e., *adjacent* realizable items combine via the rule into a realizable item over their union. Conversely, a realizable $(i, \mathrm{A}, j]$ guarantees that *some* rule $\mathrm{A} \to \mathrm{BC}$ and split index $k$ witness it, but the split is not known a priori—establishing realizability requires considering *all* candidate splits.

---

**Algorithm 1** Evaluating the realizability of an item

1. **def** R$((i, A, j])$
2.   **if** $j = i + 1$ :
3.     **return** $A \to w_j \in \mathcal{P}$
4.   **else**
5.     **guess** $a \in \{\text{SPLIT}, \text{GAP}\}$
6.     **if** $a = \text{SPLIT}$ :
7.       **guess** $A \to BC \in \mathcal{P}$
8.       **guess** $k \in (i, j)$
9.       **return** R$\big((i, B, k]\big) \wedge$ R$\big((k, C, j]\big)$
10.     **else**
11.       **guess** $Y \in \mathcal{N}$
12.       **guess** $(k, \ell) \subsetneq (i, j)$
13.       **return** R$\big((i, A, j]/(k, Y, \ell]\big) \wedge$ R$\big((k, Y, \ell]\big)$

---

**Algorithm 2** Evaluating the realizability of a slashed item

1. **def** R$((i, X, j]/(p, Y, q])$
2.   **if** $p = i \wedge q = j - 1$ :
3.     **guess** $Z \in \mathcal{N}$
4.     **return** $(X \to YZ) \in \mathcal{P} \wedge (Z \to w_j) \in \mathcal{P}$
5.   **else if** $p = i + 1 \wedge q = j$ :
6.     **guess** $Z \in \mathcal{N}$
7.     **return** $(X \to ZY) \in \mathcal{P} \wedge (Z \to w_{i+1}) \in \mathcal{P}$
8.   **else**
9.     **guess** $a \in \{\text{SPLIT}, \text{GAP}\}$
10.     **if** $a = \text{SPLIT}$ :
11.       **guess** $(X \to A B) \in \mathcal{P}$
12.       **guess** $k \in (i, j)/(p, q]$
13.       **return** $\big(\text{R}\big((i, A, k]/(p, Y, q]\big) \wedge \text{R}\big((k, B, j]\big)\big)$
        $\vee \big(\text{R}\big((i, A, k]\big) \wedge \text{R}\big((k, B, j]/(p, Y, q]\big)\big)$
14.     **else**
15.       **guess** $Z \in \mathcal{N}$
16.       **guess** $k, \ell$ with $(p, q] \subsetneq (k, \ell] \subsetneq (i, j]$
17.       **return** R$\big((i, X, j]/(k, Z, \ell]\big) \wedge$ R$\big((k, Z, \ell]/(p, Y, q]\big)$

---

We additionally use **slashed items** to record that some non-terminal already derives an inner substring of the input: the slashed item $(i, A, j]/(p, B, q]$ with $(p, q] \subsetneq (i, j]$ records that $B \stackrel{*}{\Rightarrow} \boldsymbol{w}_{(p,q]}$, and asks whether $A$ derives the sentential form obtained by replacing $\boldsymbol{w}_{(p,q]}$ inside $\boldsymbol{w}_{(i,j]}$ with the non-terminal $B$. We write $(i, A, j]/(p, B, q] = \top$, and call the slashed item **realizable** when this is the case; equivalently $A \stackrel{*}{\Rightarrow} \boldsymbol{w}_{(i,p]} B \boldsymbol{w}_{(q,j]}$. The same semantics extend to the other three bracket shapes, i.e., a slashed item of any of the forms $[i, A, j]/(p, B, q]$, $[i, A, j)/(p, B, q]$, or $(i, A, j)/(p, B, q]$ means that $A$ derives the outer yield with $\boldsymbol{w}_{(p,q]}$ excised and replaced by $B$ where the outer yield is whichever (half-)open interval the outer brackets pick out. The inner item can likewise take any of the four shapes. Analogously to $\mathcal{I}$, we denote by $\mathcal{S}$ the set of all slashed items associated with string $\boldsymbol{w}$ and a grammar $\mathcal{G}$.

A **parse tree** for $\boldsymbol{w}$ under a CFG $\mathcal{G}$ in CNF is a finite tree of items, built inductively from two deduction rules:

$$(\text{LEAF}): \quad \frac{}{(i-1, A, i]}, \qquad A \to w_i \in \mathcal{P};$$

$$(\text{COMBINE}): \quad \frac{(i, B, k] \quad (k, C, j]}{(i, A, j]}, \quad A \to B C \in \mathcal{P}.$$

The LEAF rule has no premises: whenever the grammar contains the unit production $A \to w_i$ for the input terminal $w_i$ at position $i$, it introduces the length-1 item $(i-1, A, i]$. The COMBINE rule splices two adjacent realizable items into a longer one whenever the grammar contains the binary production $A \to B C$; the split index $k$ is shared between the two premises so that the conclusion covers the union $(i, j]$ exactly, matching the calculus introduced above.

Naturally, $\boldsymbol{w} \in L(\mathcal{G})$ iff $(0, S, N]$ is realizable. A parse tree witnesses $\boldsymbol{w} \in L(\mathcal{G})$ iff its root is the item $(0, S, N]$, and an item $(i, A, j]$ is realizable iff it is the root of some sub-derivation built from these rules.

Because $\mathcal{G}$ is in CNF, any derivation $X \stackrel{*}{\Rightarrow} \boldsymbol{w}_{(i,j]}$ with $j > i + 1$ must begin with a binary rule, so any realizable item $(i, X, j]$ with $j > i + 1$ decomposes in one of two ways: *(a)* We can split at the root: there is a rule $X \to YZ$ and a split index $k \in (i, j)$ such that the items $(i, Y, k]$ and $(k, Z, j]$ are both realizable. This yields $\mathcal{O}(|\mathcal{P}| N)$ choices per item. *(b)* We can split at any internal parse-tree node: there is an item $(p, Y, q]$ such that both the slashed item $(i, X, j]/(p, Y, q]$ and the inner item $(p, Y, q]$ are realizable. Thus, there are $\mathcal{O}(|\mathcal{N}| N^2)$ choices per item. The base case $j = i + 1$ asks whether $X \to w_j$ is a rule. Slashed items decompose symmetrically, with the additional choice of which sub-derivation (left or right) brackets the assumed inner derivation, yielding $\mathcal{O}(|\mathcal{P}| N)$ root-split choices and $\mathcal{O}(|\mathcal{N}| N^2)$ internal-split choices per slashed item.

### 3.2. Parallel Parsing

The recursive characterization above couples items and slashed items: an item $(i, A, j]$ decomposes either as a root-split into two adjacent items, or as an internal-split into a slashed item and an inner item; a slashed item decomposes symmetrically into smaller slashed items and inner items. Recognizing realizability is thus naturally expressed as a pair of mutually recursive procedures—Alg. 1 recurses on items, dispatching to Alg. 2 on the slashed item produced by an internal-split; Alg. 2 recurses on slashed items, dispatching back to Alg. 1 on the inner item it spawns. Each procedure returns T iff at least one decomposition is realiz-

able; as a Boolean formula this is a disjunction $\bigvee$, ranging over candidate splits, productions, and (in Alg. 2) the choice of which child of the split inherits the inner excision.

Read as a Boolean formula, the body of Algs. 1 and 2 is a tree of $\wedge$s and $\vee$s over the predicate "the item is realizable." The conjunctions are written explicitly: each return statement combines its recursive calls with $\wedge$ and succeeds iff every branch succeeds. The disjunctions are written compactly as **guess** statements (rather than as $\bigvee_{a \in S}$). In other words, guess says pick one disjunct, and the procedure returns T iff *some* choice does. The two operators together are exactly the alternation of Chandra et al. (1981)—each **guess** is an existential ($\vee$) branch, each $\wedge$ a universal branch—so the parallel runtime is the alternating depth (Ruzzo, 1980; 1981), which equals the depth of the recursion tree: logarithmic in $N$ (cf. Lem. 3.2). By the idea just described, Algs. 1 and 2 correctly recognize membership in an arbitrary CFL[4]:

**Lemma 3.1** (Correctness). *Given a CFG $\mathcal{G}$ in CNF and $\boldsymbol{w} \in \Sigma^*$ of length $N$, $\mathrm{R}((0, \mathrm{S}, N]) = \mathrm{T}$ iff $\boldsymbol{w} \in L(\mathcal{G})$.*

We now analyze the resources required to compute $\mathrm{R}((0, \mathrm{S}, N])$ via Alg. 1 and Alg. 2, i.e., to test the membership of the input string $\boldsymbol{w}$ in $\mathcal{G}$. $\mathrm{R}$'s recursion is based on the balanced decomposition of a tree into subtrees of roughly equal size, which intuitively leads to a $\log(N)$-time procedure. Formally, we have the following fact about decomposing trees.

**Lemma 3.2** (Jordan 1869). *Let $(V, E)$ be a tree with $N \stackrel{\text{def}}{=} |V| > 1$ nodes. Then, there exists a **centroid node** $v \in V$ whose removal partitions $(V, E)$ into components, each of size at most $\lceil N/2 \rceil$.*

**Time complexity.** Lem. 3.2 allows Alg. 1 to run in a logarithmic number of recursive steps.

**Lemma 3.3** (Parallel Runtime). *Let $\boldsymbol{w} \in \Sigma^*$ be a string of length $N$ and let $\mathcal{G}$ be a context-free grammar. $\mathrm{R}((0, \mathrm{S}, N])$ terminates in at most $2\lceil \log_2(2N) \rceil + \mathcal{O}(1)$ recursive steps when $\boldsymbol{w} \in L(\mathcal{G})$.*

**Space complexity.** The bottleneck resides in computing the realizability of slashed items of the form $(i, \mathrm{X}, j]/(k, \mathrm{Y}, \ell]$, of which there are $\mathcal{O}(N^4)$. Guessing an item $(p, \mathrm{Z}, q]$ that could decompose a slashed item requires $\mathcal{O}(N^2)$ space, leading to a total space complexity of $\mathcal{O}(N^6)$.

### 3.3. Implementing the Algorithm on AHATs

Looped and padded transformers are expressively equivalent to classical models of parallel recognition (Merrill & Sabharwal, 2025; Svete et al., 2026). Combining these results with the time and space upper bounds established in

---

[4]All of §3.2's proofs figure in §B.1.

§3.2, we can show (see §B.3 for the proof):

**Theorem 3.1.** $\mathrm{CFL} \subseteq \mathrm{MAHAT}_6^1 \subseteq \mathrm{AHAT}_7^1$.

## 4. Parallel Recognition of Unambiguous CFLs

Thm. 3.1 shows that $\log(N)$-deep MAHATs with $\mathcal{O}(N^6)$ padding can recognize all CFLs. Intuitively, the padding in Thm. 3.1 handles ambiguity in a general CFL by storing all the ways in which we can decompose an item. Thus, guessing how to decompose an item requires a substantial amount of space, and one might expect that *unambiguous* grammars would require less space for recognition. Accordingly, we next study **unambiguous** CFLs (UCFLs), which admit at most one derivation for any string.

Unambiguity is a natural and exploitable property. For example, programming language parsers such as LR parsers rely on deterministic (therefore unambiguous) CFLs to process inputs in linear time. Moreover, transformers struggle to parse ambiguous grammars (Khalighinejad et al., 2023) and struggle to process syntactically ambiguous natural language sentences (Liu et al., 2023a). In this section, we show that UCFLs can be recognized by transformers with less padding than arbitrary CFLs if slightly more depth overhead is allowed:

**Theorem 4.1.** $\mathrm{UCFL} \subseteq \mathrm{MAHAT}_3^2 \subseteq \mathrm{AHAT}_4^2$.

We do so by first formalizing Chytil et al.'s (1991) unambiguous CFL recognition algorithm with a smaller space complexity in $\log^2(N)$-time. We then translate this algorithm into AHATs with a tractable amount of padding.

### 4.1. Parsing Unambiguous CFLs

Our goal is again to decide whether $\boldsymbol{w} \in L(\mathcal{G})$ by computing the realizability of $(0, \mathrm{S}, N]$. In contrast to Algs. 1 and 2, which start from the goal item $(0, \mathrm{S}, N]$ and recurse downward, here we instead work bottom-up and *in parallel across all items*: at every iteration $t$, the algorithm computes realizability for *every* item $\iota$ simultaneously, regardless of whether that item happens to lie on a derivation of $(0, \mathrm{S}, N]$. Concretely, the algorithm maintains a growing set $\mathcal{I}_0 \subseteq \mathcal{I}_1 \subseteq \cdots$ of items already certified as realizable, and at each iteration extends it by one "CKY-style" combination step composed with a reachability closure. Unambiguity specifically lets us evaluate this closure in $\mathcal{O}(\log N)$ parallel depth (Fact 4.1).

First, we define the initial set of items

$$\mathcal{I}_0 \stackrel{\text{def}}{=} \{(i-1, \mathrm{A}, i] \mid \mathrm{A} \to w_i \in \mathcal{P}\}, \tag{1}$$

i.e., length-1 items witnessed by the unit productions of $\mathcal{G}$. These are the base-case items certifiable as realizable

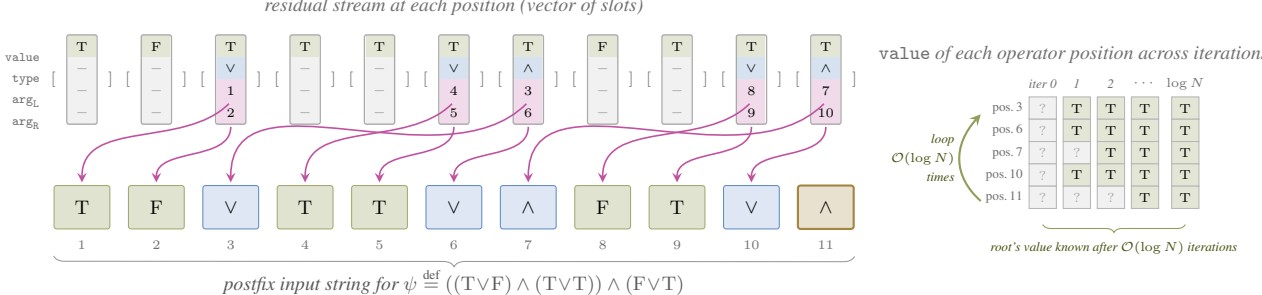

*Figure 1.* **Postfix encoding of a variable-free Boolean formula $\psi$ used by Lem. 4.1.** The example formula is $\psi \overset{\text{def}}{=} ((\text{T} \vee \text{F}) \wedge (\text{T} \vee \text{T})) \wedge (\text{F} \vee \text{T})$, whose postfix serialization is T F $\vee$ T T $\vee$ $\wedge$ F T $\vee$ $\wedge$. No padding symbols are appended: every position of the input is either a leaf value (T/F) or a binary operator ($\wedge/\vee$). At each position the residual stream carries the slots [value, type, $\arg_{\text{L}}$, $\arg_{\text{R}}$]: at leaf positions only value is set (to the input value); at operator positions type encodes the connective and $\arg_{\text{L}}, \arg_{\text{R}}$ are pointers to the two operand positions (purple arrows)—determined directly from postfix structure by the C-RASP preprocessing of Cor. 4.1. The right panel shows that value at each operator position is filled *iteratively*: at every loop iteration, operator positions whose two operands' values are already known pebble up their own value, so after $\mathcal{O}(\log N)$ iterations the root's value is known.

directly from the input. Then, we define the relation

$$\big((i, \text{B}, k], (k, \text{C}, j], (i, \text{A}, j]\big) \in \mathcal{R} \iff \text{A} \to \text{BC} \in \mathcal{P}. \tag{2}$$

$\mathcal{R}$ encodes a single COMBINE step of the parse-tree calculus (§3.1): if B derives $\boldsymbol{w}_{(i,k]}$ and C derives the adjacent right span $\boldsymbol{w}_{(k,j]}$ and the grammar contains $\text{A} \to \text{BC}$, then A derives the union $\boldsymbol{w}_{(i,j]}$. In other words, $\mathcal{R}(\iota_1, \iota_2, \iota_3)$ holds iff items $\iota_1$ and $\iota_2$ are the two CKY antecedents that justify $\iota_3$. Given $\mathcal{R}$, the algorithm maintains a growing set of **marked** items, where "$\iota$ marked at step $t$" is the algorithm's running certificate that $\iota$ is realizable, established by iteration $t$. The base items $\mathcal{I}_0$ are marked at step 0. The marking sets will satisfy $\mathcal{I}_0 \subseteq \mathcal{I}_1 \subseteq \cdots \subseteq \mathcal{I}_* \subseteq \mathcal{I}$, and we will see below that the fixpoint $\mathcal{I}_*$ contains exactly the realizable items, so testing $(0, \text{S}, N] \in \mathcal{I}_*$ decides membership in $L(\mathcal{G})$. We extend $\mathcal{I}_t$ inductively; assuming the marked set $\mathcal{I}_{t-1}$ at iteration $t-1$ is available, we build $\mathcal{I}_t$ in two steps: (*1*) Package every CKY combination that uses a witness from $\mathcal{I}_{t-1}$ into a directed graph and (*2*) Chain arbitrarily many such combinations together within the same iteration, so that $\mathcal{I}_t$ contains items realizable by a derivation that bottoms out in $\mathcal{I}_{t-1}$ via any chain of new combinations.

Concretely, we construct a graph $(\mathcal{I}, \mathcal{E}_t)$ whose edges record one-step implications between items, conditional on $\mathcal{I}_{t-1}$:

$$\mathcal{E}_t \overset{\text{def}}{=} \big\{ \iota_3 \to \iota_1 \mid \iota_3 \notin \mathcal{I}_{t-1} \text{ and } (\mathcal{R}(\iota_1, \iota_2, \iota_3) \text{ or} \tag{3}$$
$$\mathcal{R}(\iota_2, \iota_1, \iota_3)) \text{ for some } \iota_2 \in \mathcal{I}_{t-1} \big\}.$$

Intuitively, the edge $\iota_3 \to \iota_1$ encodes that *if $\iota_1$ is realizable, then so is $\iota_3$*, where the witness $\iota_2$ is drawn from the items already marked at step $t-1$. We write $\text{DG}(\mathcal{I}) \overset{\text{def}}{=} (\mathcal{I}, \mathcal{E}(\mathcal{I}))$ for the dependency graph obtained by setting $\mathcal{I}_{t-1} = \mathcal{I}$ in Eq. (3), so that $\text{DG}(\mathcal{I}_{t-1}) = (\mathcal{I}, \mathcal{E}_t)$ and, in particular, $\text{DG}(\mathcal{I}_0)$ is the *initial* dependency graph $(\mathcal{I}, \mathcal{E}_1)$. Writing $\text{R}_t(\iota)$ for the set of items reachable from $\iota$ in $(\mathcal{I}, \mathcal{E}_t)$

$(|\text{R}_t(\iota)| \leq |\mathcal{I}| = \mathcal{O}(N^2))$, we mark $\iota$ at step $t$ exactly when it reaches some item already marked at step $t-1$:

$$\iota \in \mathcal{I}_t \iff \text{R}_t(\iota) \cap \mathcal{I}_{t-1} \neq \emptyset. \tag{4}$$

In particular, $\mathcal{I}_{t-1} \subseteq \mathcal{I}_t$. The algorithm we implement on transformers in §4.2 stages this as an outer loop over $t$: at each $t$ it *(i)* computes the new edge set $\mathcal{E}_t$ from the previously marked set $\mathcal{I}_{t-1}$ by a single CKY-style probe of $\mathcal{R}$, and *(ii)* evaluates the reachability test of Eq. (4) for every item in parallel to obtain $\mathcal{I}_t$. The outer loop terminates once $\mathcal{I}_t = \mathcal{I}_{t-1}$, at which point no new combinations are possible and we recover $\mathcal{I}_*$ as the set of all *realizable* items. Chytil et al. (1991) show that at most $\mathcal{O}(\log N)$ iterations are required until the procedure reaches a fixed point. To bound the total runtime of the procedure, it remains to bound the complexity of each reachability problem. Chytil et al. (1991) again show that in the case of an unambiguous CFG, each graph reachability problem takes at most $\mathcal{O}(\log N)$ time. The basis is the following fact.

**Fact 4.1** (Chytil et al. (1991)). *Let $\mathcal{G}$ be an unambiguous CFG and $t \geq 1$. For any pair of items $\iota, \iota' \in \mathcal{I}$ in the graph $(\mathcal{I}, \mathcal{E}_t)$, there is at most one directed path from $\iota$ to $\iota'$.*

Thus, $\text{R}_t(\iota)$ is a *tree* rooted at $\iota$ for every item $\iota \in \mathcal{I}$. The marking condition Eq. (4) can then be expressed recursively along this tree: $\iota \in \mathcal{I}_t$ holds iff some node of $\text{R}_t(\iota)$ lies in $\mathcal{I}_{t-1}$, which we can unfold edge by edge as

$$\mathbb{1}\{\iota \in \mathcal{I}_t\} \overset{\text{def}}{=} \mathbb{1}\{\iota \in \mathcal{I}_{t-1}\} \vee \bigvee_{\iota \to \iota' \in \mathcal{E}_t} \mathbb{1}\{\iota' \in \mathcal{I}_t\}, \tag{5}$$

The recursion terminates at the leaves of $\text{R}_t(\iota)$—items with no outgoing edge in $\mathcal{E}_t$—since by Eq. (3) any item with an outgoing edge lies outside $\mathcal{I}_{t-1}$, so the two disjuncts in Eq. (5) are mutually exclusive. Unrolled along the tree

$R_t(\iota)$, this collapses to the flat Boolean disjunction

$$\mathbb{1}\left\{\iota \in \mathcal{I}_t\right\} \iff \bigvee_{\iota' \in R_t(\iota)} \mathbb{1}\left\{\iota' \in \mathcal{I}_{t-1}\right\}. \tag{6}$$

Each internal node of $R_t(\iota)$ is an $\vee$ gate and each leaf is the indicator $\mathbb{1}\left\{\iota' \in \mathcal{I}_{t-1}\right\}$, so the formula is evaluable in $\mathcal{O}(\log N)$ parallel depth (Rytter, 1985). Putting the pieces together, one outer iteration of the recognition algorithm consists of *(1)* building the dependency graph $\mathcal{E}_t$ from $\mathcal{I}_{t-1}$ by a single grammar-rule lookup per candidate triple, and *(2)* evaluating, in parallel for every item $\iota$, the Boolean formula Eq. (6) over the tree $R_t(\iota)$. This is the recipe we follow in §4.2: each step is implemented by a fixed transformer block, the inner reachability tree by Lem. 4.1, and the outer loop by $\mathcal{O}(\log N)$ loop iterations—yielding the $\log^2$-depth recognizer of Thm. 4.1.

### 4.2. Unambiguous CFLs on Transformers

This section outlines the proof of Thm. 4.1 in two steps:[5] *(1)* in Lem. 4.1, we first show how transformers can compute the value of variable-free Boolean formulas in $\mathcal{O}(\log(N))$ parallel depth—this gives us the parallel evaluator we need for the reachability formula Eq. (6)—and then *(2)* we show how to implement the unambiguous CFL recognition algorithm on transformers by using the first step as a subroutine to evaluate Eq. (6) at every item in parallel.

**Lemma 4.1.** *For any variable-free postfix Boolean formula $\psi$ on $N$ inputs, there exists a $\mathcal{O}(\log(N))$-looped unpadded transformer that computes $\psi$'s value.*

*Proof intuition.* We feed $\psi$ in postfix form, so every position of the input is either a leaf value (T/F) or a binary operator ($\wedge/\vee$) and no padding is appended. A fixed-depth C-RASP preprocessing pass (Lem. B.2) populates the residual stream at each operator position with the connective and with pointers $\text{arg}_\text{L}$, $\text{arg}_\text{R}$ to its two operand positions, which can be read off directly from the postfix structure. This puts the nodes of $\psi$'s binary expression tree in bijection with the input positions, so each node owns its own residual stream and no padding is needed to host intermediate values. On top of this representation the transformer runs Rytter's (1985) parallel pebble game: at each loop iteration, every operator position whose two operands' values are already known reads them via $\text{arg}_\text{L}$, $\text{arg}_\text{R}$ and pebbles up its own value value. Since the expression tree has depth $\mathcal{O}(\log(N))$ in the balanced case and Rytter's pebble game halves the unresolved depth at every iteration in general, after $\mathcal{O}(\log(N))$ iterations every operator position—in particular the root—has its value filled in, and the transformer reads off $\psi$'s value from the root position. Fig. 1 illustrates the postfix

---

[5] See §B.4 for the full proofs.

encoding and the iterative filling of value on the example $\psi \stackrel{\text{def}}{=} ((T\vee F) \wedge (T\vee T)) \wedge (F\vee T)$. ∎

Lem. 4.1 already yields a result of independent interest. We denote by **BFVP** the set of variable-free Boolean formulas that evaluate to T—the *Boolean formula value problem* of Buss (1987), the canonical $\text{NC}^1$-complete language. BFVP is an unambiguous CFL whose recognition *requires* $\mathcal{O}(\log(N))$-time on classical models of parallel computation: any shallower parallel model (in particular $\text{TC}^0$, and hence fixed-depth transformers) cannot recognize BFVP unless $\text{TC}^0 = \text{NC}^1$. Applying Lem. 4.1 to the indicator of acceptance gives the following.

**Corollary 4.1.** $\text{BFVP} \in \text{AHAT}_0^1$.

Precisely, Lem. B.2 shows how to correctly encode the arguments of each operator in an input Boolean formula in postfix notation. Lem. B.3 then shows how to correctly evaluate a formula whose binary expression tree is properly encoded in a transformer—a gadget we leverage to show how transformers can recognize UCFLs with $\mathcal{O}(\log^2(N))$ looping layers and $\mathcal{O}(N^3)$ padding.

**Theorem 4.1.** $\text{UCFL} \subseteq \text{MAHAT}_3^2 \subseteq \text{AHAT}_4^2$.

*Proof intuition.* The construction implements the recognition procedure of §4.1 as an MAHAT, maintaining the marked set $\mathcal{I}_t$ across $\mathcal{O}(\log N)$ outer iterations. Each item $(i, \text{A}, j]$ is assigned a padding symbol that stores a three-valued realizability bit in $\{\text{T}, \text{F}, \bot\}$ (which can be represented in a transformer by three distinct integers) tracking whether $(i, \text{A}, j] \in \mathcal{I}_t$; there are $\mathcal{O}(N^2)$ such item-paddings. For every candidate edge in Eq. (3)—of which there are $\mathcal{O}(N^3)$, summed over the right-witness pairs $(i, \text{A}, j], (i, \text{B}, k]$ and their left-witness counterparts—we allocate one edge-padding symbol.

Before the outer loop, a constant-depth block initializes the item-paddings for the length-1 items $(i - 1, \text{A}, i]$ to T or F according to whether $\text{A} \to w_i \in \mathcal{P}$, realizing $\mathcal{I}_0$; all other item-paddings start at $\bot$.

Each outer iteration $t$ then runs four constant-depth blocks that implement Eqs. (3) and (4). *(1) Build $\mathcal{E}_t$.* Each edge-padding for a pair $(i, \text{A}, j], (i, \text{B}, k]$ enumerates the $\mathcal{O}(|\mathcal{N}|)$ candidate witnesses $(k, \text{C}, j]$ with $\text{A} \to \text{BC} \in \mathcal{P}$ via a feed-forward network, attends to their item-paddings to read off realizability, and flips its edge bit on when some witness lies in $\mathcal{I}_{t-1}$; the left-witness family is constructed symmetrically. *(2) Binarize.* By unambiguity (Fact 4.1), the reachable subgraph $R_t(\iota)$ from each item $\iota$ is an arborescence, so reachability on $(\mathcal{I}, \mathcal{E}_t)$ can be reduced to evaluating a Boolean formula on a binary tree (Chytil et al., 1991). We apply a constant-depth graph transform $\mathcal{T}$ that replaces every fan-out of size $M \geq 3$ at an item $\iota$ by a right-deep chain

of $M - 2$ fresh intermediary padding symbols indexed by $(\iota, i)$, yielding a binary graph with the same reachability structure; the $(\iota, i)$ encoding and the resulting $\texttt{arg}_\texttt{L}$, $\texttt{arg}_\texttt{R}$ pointers are then computed and added to the residual stream. *(3) Evaluate reachability.* Treating every internal node of the binarized graph as an $\vee$-gate and every leaf as T iff the underlying item is in $\mathcal{I}_{t-1}$, a single parallel pebble-game pass (Lem. B.3) evaluates the formula $\bigvee_{\iota' \in \texttt{R}_t(\iota)} \mathbb{1}\{\iota' \in \mathcal{I}_{t-1}\}$ at every item $\iota$ simultaneously in $\mathcal{O}(\log N)$ layers. *(4) Write back.* Each item-padding reads its own root value and updates its realizability bit, implementing Eq. (4).

After $\mathcal{O}(\log N)$ outer iterations $\mathcal{I}_t$ reaches its fixpoint $\mathcal{I}_*$ (Chytil et al., 1991), and $\mathbb{1}\{\boldsymbol{w} \in L(\mathcal{G})\}$ can be read off the item-padding for $(0, \texttt{S}, N]$. The total depth is $\mathcal{O}(\log N) \cdot \mathcal{O}(\log N) = \mathcal{O}(\log^2(N))$, and the padding budget is $\mathcal{O}(N^3)$, dominated by the per-edge and binarization-intermediary symbols. ∎

### 4.3. Parsing Linear Unambiguous CFLs

Finally, we show how *linearity* further reduces the resources needed to recognize unambiguous CFLs. A **linear** CFL is one recognized by a CFG where each rule is of the form $\texttt{A} \to a\texttt{B}$, $\texttt{A} \to \texttt{B}a$, or $\texttt{A} \to a$. While restricted, linear CFLs capture a wide range of features of context-freeness. For example, *balanced counting* can be modeled by the linear CFL $\{a^N b^N \mid N \geq 0\}$, and *symmetry* can be modeled by the linear CFL $\{\boldsymbol{w}\overleftarrow{\boldsymbol{w}} \mid \boldsymbol{w} \in \Sigma^*\}$.

We consider unambiguous linear[6] CFLs (ULCFLs). Linearity simplifies the dependency graphs $(\mathcal{I}, \mathcal{E}_t)$ of Eq. (3): solving an item $\iota = (i, \texttt{A}, j] \in \mathcal{I}$ via a rule $\texttt{A} \to a\texttt{B}$ reduces to the single sub-item $(i+1, \texttt{B}, j]$ (or symmetrically $(i, \texttt{B}, j-1]$), so each item has only a *constant* number of outgoing edges in $\mathcal{E}_t$ proportional to $|\mathcal{P}|$. Moreover, because each binary combination $\mathcal{R}(\iota_1, \iota_2, \iota_3)$ in a linear grammar pairs one non-terminal sub-item with a length-1 terminal sibling, every witness $\iota_2$ required by Eq. (3) already lies in $\mathcal{I}_0$. Hence every edge that ever appears in any $\mathcal{E}_t$ is already present in $\mathcal{E}_1$, and reachability to $\mathcal{I}_0$ in $(\mathcal{I}, \mathcal{E}_1)$ certifies realizability directly:

**Proposition 4.1.** *If $L \in$ ULCFL, then the growing set of items $\mathcal{I}_1, \ldots, \mathcal{I}_*$ satisfy $\mathcal{I}_1 = \mathcal{I}_*$.*

Therefore, a single reachability pass suffices and no outer iteration is needed.

### 4.4. Linear Unambiguous CFLs on Transformers

Combining Prop. 4.1 with Thm. 4.1 reduces both the time and space complexities:

---

[6]A CFL can be induced by both a non-linear unambiguous grammar and by a different linear, ambiguous grammar. We consider grammars that are *simultaneously* linear and unambiguous.

**Theorem 4.2.** $\text{ULCFL} \subseteq \text{MAHAT}_2^1 \subseteq \text{AHAT}_3^1$.

See §B.4 for full proofs.

## 5. Experiments

We conduct experiments to elicit the impact of looping and padding when recognizing CFLs, and provide more details on our experimental setup in §C. We train transformer classifiers on CFLs of varying degrees of complexity.

- **Balanced Counting**: the language of strings with some number of $a$s followed by the same number of $b$s, defined as $L = \{a^N b^N \mid N \geq 0\}$ with $\Sigma = \{a, b\}$. This CFL is linear and deterministic. Log-precision transformers succeed theoretically (Yang & Chiang, 2024) and empirically (Bhattamishra et al., 2020) on this CFL.
- **Dyck**: the language of nested strings of parentheses of $k$ types, which we denote by $\text{Dyck}(k)$. We consider $\text{Dyck}(1)$ and $\text{Dyck}(2)$. This CFL is non-linear and deterministic. Fixed-depth transformers with log-precision can recognize $\text{Dyck}(k)$ for any $k$ (Hayakawa & Sato, 2024). Transformers empirically succeed on $\text{Dyck}(1)$ (Bhattamishra et al., 2020) but struggle on Dyck languages with more than 1 bracket type (Ebrahimi et al., 2020).
- **Palindrome**: the language $L = \{\boldsymbol{w}\overleftarrow{\boldsymbol{w}} \mid \boldsymbol{w} \in \Sigma^*\}$ for some alphabet $\Sigma$. We focus on a binary alphabet. This CFL is linear unambiguous and non-deterministic. Fixed-depth transformers with hard attention and unbounded precision can recognize this language (Hao et al., 2022). Empirically, transformers struggle to recognize Palindrome (Butoi et al., 2025).
- **Boolean formula value problem** (BFVP): the set of variable-free Boolean formulas that evaluate to T. We consider formulas in both the standard *infix* notation (e.g., $1 \vee 0$ is in infix notation) and *postfix* notation (e.g., $1\ 0\ \vee$ is in postfix notation). Parallel algorithms for BFVP typically rely on postfix notation (Buss, 1987; Buss et al., 1992). We have proven (Cor. 4.1) that log-depth and no padding suffice to recognize this language in postfix notation with causally-masked transformers.

These languages vary in complexity, allowing us to test transformers' ability to learn CFL recognition constructions when they are extended with looping and padding. In particular, Balanced Counting and $\text{Dyck}(1)$ are in **C-RASP** — a class that describes what languages transformers should generalize on (Yang & Chiang, 2024; Huang et al., 2025b). On the other hand, $\text{Dyck}(2)$, Palindrome and BFVP are not in C-RASP. Importantly, BFVP requires growing depth (i.e., log-depth), assuming $\text{TC}^0 \neq \text{NC}^1$. Cor. 4.1 predicts that BFVP in postfix notation should be expressible with logarithmic looping and no padding. Our results are presented in Tab. 2.

| Language | Fixed-depth transformer | $\mathcal{O}(\log(N))$ looping | $\mathcal{O}(\log(N))$ looping and $\mathcal{O}(N)$ padding |
|---|---|---|---|
| Balanced Counting | 90 | **94** | 93 |
| Dyck(1) | 85 | **86** | **86** |
| Dyck(2) | 83 | 84 | **87** |
| Palindrome | 68 | 67 | **72** |
| BFVP (infix) | 80 | 78 | **81** |
| BFVP (postfix) | 67 | **75** | 75 |

*Table 2.* Maximum accuracy (out of 100) across 5 random seeds of different transformer variants (fixed-depth, $\log(N)$ looped and unpadded, $\log(N)$ looped and $\mathcal{O}(N)$ padded) across a range of context-free languages. The highest score per row are highlighted.

**Results.** For Balanced Counting—a C-RASP language on which fixed-depth transformers already achieve 90% accuracy, we find that logarithmic looping increases performance by 4%. On the other hand, the performance gap between fixed- and looped-transformers on the C-RASP language Dyck(1) is negligible. Interestingly, there is a larger performance discrepancy between looped, padded transformers and fixed-depth ones on Dyck(2)—a language outside of C-RASP (Hu et al., 2025). For Palindrome, looping and padding together outperform both the vanilla and looping-only baseline by 4%. For the infix variant of BFVP, looped transformers slightly underperform the fixed-depth baseline, and looped, padded transformers slightly overperform. However, we notice the largest accuracy gains in our experiments on *postfix* BFVP: looped transformers (with or without padding) outperform fixed-depth ones by 8%. This result agrees with Cor. 4.1, which states that looped transformers without padding can express postfix BFVP. Overall, we find that looping and padding offer moderate performance gains over fixed-depth transformers on a range of CFLs, with the largest gains on the theoretically motivated postfix BFVP language.

## 6. Discussion

**Transformers and parallel parsing.** Our work provides a theoretical framework for understanding how transformers can internally process syntax by formulating parsing as a *parallel* procedure implementable by looped- and padded-transformers (§3, §4). Interestingly, transformers *in practice* seem to implement some form of parallel parsing. Schulz et al. (2025) show that transformers parse by learning sub-grammars—grammars that generate *substrings* of the original grammar—in parallel. Allen-Zhu & Li (2025) show via probing that transformers simulate a dynamic program that manipulates items of the form $[i, \mathrm{X}, j]$ and that they implement *memory reads* across positions to combine the solutions of items. While they state that such an algorithm can be naively implemented in polynomial time, our constructions leverage transformers' inherent parallelism to show it can be implemented exponentially faster (i.e., in logarithmic time). Finally, Zhao et al. (2023) show with probing that transformers encode the syntactic information relevant for the *Inside-outside algorithm*, which computes string probabilities under *probabilistic* CFGs. Interestingly, the Inside-outside algorithm can also be formulated as the computation of the weights of items and slashed items. Altogether, while several interpretability results hint at which syntactic information may be used by transformers for parsing, Thm. 3.1 unifies these results by providing an exact construction of how looped transformers can exactly implement parallel parsing.

**Learnability.** Expressivity alone does not capture transformers' empirical capabilities. A function that is expressible may not be easily learnable (Hahn & Rofin, 2024), and the gap between expressivity and learnability has been documented empirically for both recognition (Butoi et al., 2025) and probabilistic modeling of formal languages (Borenstein et al., 2024). Although our work provides a framework for understanding how transformers can express a CFL, better understanding the ability of a transformer to *learn* a CFL through training dynamics (Huang et al., 2025a), loss landscape analysis (Hahn & Rofin, 2024), or scaling laws (Merrill et al., 2026) could provide more complete understanding of how transformers process syntax.

## 7. Conclusion

While it was known that logarithmic-depth transformers could recognize regular languages via parallel simulation of automata (Liu et al., 2023b; Li et al., 2025), our results generalize this to show that logarithmic depth suffices for recognizing CFLs as well in the presence of padding symbols. This means that transformers can recognize an arbitrary CFL with exponentially faster sequential runtime compared to serial parsing algorithms such as CKY. However, such a parallel procedure requires a large space overhead of padding symbols. Fortunately, we show that grammar properties such as unambiguity can reduce the amount of space required for recognition. We also show that unpadded transformers with logarithmic looping can evaluate any Boolean formula. Empirically, we found that looping and padding provide modest benefits to transformers on certain CFL recognition tasks, leaving open a more extensive empirical study across CFLs and padding budgets to future work.

## Impact Statement

This paper presents work whose goal is to advance the field of machine learning. There are many potential societal consequences of our work, none of which we feel must be specifically highlighted here.

## Acknowledgments

We thank attendees of the Formal Languages and Neural Networks (FLaNN) seminar for insightful discussions about this work. Anej Svete is supported by an ETH AI Center Doctoral Fellowship. William Merrill was supported by a Two Sigma PhD fellowship, an NSF Graduate Research Fellowship, and the Allen Institute for AI. We used generative AI to improve our writing. Every modification introduced by generative AI was carefully reviewed by the authors, who take full responsibility for it.

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

# A. Transformer Models: Details

We restate our definition of a transformer, and formalize each of its components.

**Definition 2.2** (Averaging hard-attention transformer). *An **averaging hard-attention transformer** (AHAT) $T$ of width $D$ and depth $L$ is a tuple $T = (\text{emb}, (\mathcal{L}^{(\ell)})_{\ell=1}^{L}, c)$ where:*

- *$\text{emb} \colon \Sigma^* \to (\mathbb{Q}^D)^*$ is a position-wise **input embedding** that maps each input symbol to a vector in $\mathbb{Q}^D$;*
- *each **layer** $\mathcal{L}^{(\ell)} \colon (\mathbb{Q}^D)^* \to (\mathbb{Q}^D)^*$ is a length-preserving map that composes a layer-normalization, an attention sublayer whose attention weights are computed by the averaging hard-attention projection $\text{hardmax}$, and a position-wise feedforward network;*
- *$c \colon \mathbb{Q}^D \to \{0, 1\}$ is a **classification function**.*

*The transformer first computes the length-preserving mapping $\mathcal{L}^{(L)} \circ \cdots \circ \mathcal{L}^{(1)} \circ \text{emb}(\boldsymbol{w})$. Let $\boldsymbol{x}_{\text{EOS}}^L \in \mathbb{Q}^D$ be the final contextual representation of $\text{EOS}$ after the $L$ transformer layers. We say the transformer accepts $\boldsymbol{w}$ iff $T(\boldsymbol{w}) \overset{\text{def}}{=} c(\boldsymbol{x}_{\text{EOS}}^L) = 1$.*

**Marker positions.** We recall that the input string is augmented with both the beginning-of-sequence (BOS) and end-of-sequence (EOS) symbols. We leverage these positions as *marker* positions. BOS can be leveraged as an anchor position that every other position can always attend to, which for instance is useful to compute position encodings (Merrill & Sabharwal, 2024b). On the other hand, because EOS can attend to any string position throughout the forward pass regardless of the masking used, it is common to use EOS's final representation to classify a string.

**Embedding layer.** The **input embedding** $\text{emb} \colon \Sigma^* \to (\mathbb{Q}^D)^*$ applies an injective position-wise embedding function to each symbol in the input string $\boldsymbol{w}$, and is therefore a length-preserving function. For example emb could be a non-parametrized function such as a one-hot embedding, or could apply a linear mapping $\Sigma \to \mathbb{Q}^D$ via a weight matrix $\boldsymbol{W} \in \mathbb{Q}^{|\Sigma| \times D}$ to every symbol. We use the latter in our implementation of the transformer in §5.

**Transformer layers.** $\mathcal{L}^{(\ell)}$ for $\ell \in (0, L]$ denotes a **transformer layer**—a mapping $\mathcal{L}^{(\ell)} \colon (\mathbb{Q}^D)^* \to (\mathbb{Q}^D)^*$ that updates the symbol representations. The components of a transformer layer are the **layer normalization** LN, the **attention layer** $\boldsymbol{f}_{\text{att}}^{(\ell)}$ and the **feedforward network** $\mathbf{F}^{(\ell)}$. Concretely:

$$\mathcal{L}^{(\ell)} \overset{\text{def}}{=} \mathbf{F}^{(\ell)} \circ \boldsymbol{f}_{\text{att}}^{(\ell)} \circ \text{LN}^{(\ell)} \tag{7}$$

We recall layer-normalization maps a vector $\boldsymbol{x} \in \mathbb{Q}^N$ of some dimension $N$ to $\frac{\boldsymbol{x}'}{\|\boldsymbol{x}'\|}$ where $\boldsymbol{x}' \overset{\text{def}}{=} \boldsymbol{x} - \frac{\sum_{x_i \in \boldsymbol{x}} x_i}{N}$. We assume **multi-pre-norm** (Merrill & Sabharwal, 2024b). In standard pre-norm, we apply a layer-normalization to the entire hidden state of each symbol. With multi-pre-norm, we allow each sublayer to take $k$ different projections of its input, apply layer-norm to each and concatenate. Crucially, multi-pre-norm allows us to partition the hidden state and normalize disjoint subsets thereof, which we will rely on in our proofs.

$\mathbf{F}^{(\ell)} \colon (\mathbb{Q}^D)^* \to (\mathbb{Q}^D)^*$ is a position-wise function that applies the same feedforward network to every symbol of the sequence. It is parametrized by weight matrices of the form $\boldsymbol{W} \in \mathbb{Q}^{m \times D}$ and $\boldsymbol{U} \in \mathbb{Q}^{D \times m}$. A feedforward network $\mathbf{F}^{(\ell)}$ can nest functions of the form $\boldsymbol{U} \text{ReLU}(\boldsymbol{W}\boldsymbol{z})$ where $\boldsymbol{z} \in \mathbb{Q}^D$ is an intermediate value.

The **attention mechanism** is defined by the function $\boldsymbol{f}_{\text{att}}^{(\ell)} \colon (\mathbb{Q}^D)^* \to (\mathbb{Q}^D)^*$. We denote by $\boldsymbol{k}_i^{(\ell)}, \boldsymbol{q}_i^{(\ell)}, \boldsymbol{v}_i^{(\ell)}$ the key, query and value vectors, respectively, for symbol $i$ at layer $\ell$. $\boldsymbol{f}_{\text{att}}^{(\ell)}$ is defined as follows:

$$\boldsymbol{f}_{\text{att}}^{(\ell)}((x_1, \cdots, x_T)) \overset{\text{def}}{=} (y_1, \cdots, y_T) \tag{8a}$$

$$y_i \overset{\text{def}}{=} x_i + \sum_{i' \in m(i)} s_{i'} \boldsymbol{v}_{i'}^{(\ell)} \tag{8b}$$

$$s = \text{proj}(\{\text{score}(\boldsymbol{k}_{i'}^{(\ell)}, \boldsymbol{q}_i^{(\ell)})\}) \tag{8c}$$

$m(i)$ is a set that defines the **masking** used by the transformer. For instance, $m(i) = \{i' \mid i' < i\}$ refers to strict causal masking and $m(i) = (0, N]$ refers to no masking. score is a scoring function that maps two vectors of the same size to a scalar. Typically, the dot-product score is used with $\text{score}(x_1, x_2) \overset{\text{def}}{=} \langle x_1, x_2 \rangle$.

Throughout layers, the hidden state $y_i$ of a symbol at position $i$ continuously evolves as it cumulatively adds up the outputs of the attention mechanism. We call this cumulative sum $y_i$ over layers the **residual stream** at $i$.

proj is a projection function that normalizes the scores into weights for the symbol values. Following previous work, we assume an **averaging hard attention** transformer (AHAT), which concentrates the attention weights on the symbols that maximize the attention score (Merrill et al., 2022; Strobl, 2023). Formally, we have $\texttt{proj} = \mathrm{hardmax}$:

**Definition A.1.** *Averaging hard attention is computed with the* hardmax *projection function:*

$$\mathrm{hardmax}\left(\boldsymbol{x}\right)_d \overset{\text{def}}{=} \begin{cases} \frac{1}{m} & \textbf{if } d \in \mathrm{argmax}\left(\boldsymbol{x}\right) \\ 0 & \textbf{otherwise} \end{cases} \tag{9}$$

*for $d \in (0, D]$, where $\boldsymbol{x} \in \mathbb{Q}^D$ and $m \overset{\text{def}}{=} |\mathrm{argmax}\left(\boldsymbol{x}\right)|$ is the cardinality of the argmax set.*

**Language recognition.** The transformer layers are vector-valued functions by definition. To treat a transformer as a language recognizer of signature $\Sigma^* \to \{0, 1\}$, we use the final representations computed by a transformer for binary classification of strings. We denote by $\boldsymbol{x}_{\text{EOS}}^L$ the hidden state of EOS after passing it through the $L$ transformer layers. Typically, string recognition is based on $\boldsymbol{x}_{\text{EOS}}^L$ as EOS is the only symbol that can attend to every other position, regardless of the masking used. This allows us to define a transformer's language based on a linear classifier $c$.

$$\mathsf{T}(\boldsymbol{w}) = c(\boldsymbol{x}_{\text{EOS}}^L) \overset{\text{def}}{=} \mathbb{1}\left\{\sigma(\boldsymbol{\theta}^\top \boldsymbol{x}_{\text{EOS}}^L) > 0.5\right\} \tag{10}$$

Where $\sigma$ is the sigmoid activation function.

**Precision.** Following previous work (Merrill & Sabharwal, 2025; 2024b; 2023a), we assume $\log$-precision transformers, i.e., we allow the transformer to manipulate values that can be represented with $\mathcal{O}(\log(N))$ bits for an input of length $N$. It is a minimally extended idealization that enables the transformer to store indices and perform sums over an unbounded number of symbols, two crucial capabilities for our constructions.

**Layer-norm hash.** We will often use the **layer-norm hash** building block (Merrill & Sabharwal, 2024b). It is particularly useful for equality checks between values across different symbols, especially with a potentially unbounded number of queries and keys.

**Definition A.2** (Merrill & Sabharwal, 2024b). *Given a scalar $z \in \mathbb{R}$, its **layer-norm hash** is $\phi(z) \overset{\text{def}}{=} \langle z, 1, -z, -1 \rangle / \sqrt{z^2 + 1}$.*

Layer-norm hash is scale invariant, and $\phi(q) \cdot \phi(k) = 1$ iff $q = k$. In other words, the inner product of scalars $q$ and $k$, even if computed at different positions $i$ and $j$, respectively, allows us to check for the equality of $q$ and $k$. Layer-norm hash therefore allows us to perform equality checks over elements of residual streams at different positions.

## B. Proofs

### B.1. Parallel Parsing

**Lemma 3.1** (Correctness). *Given a CFG $\mathcal{G}$ in CNF and $\boldsymbol{w} \in \Sigma^*$ of length $N$, $\mathrm{R}((0, \mathrm{S}, N]) = \mathsf{T}$ iff $\boldsymbol{w} \in L(\mathcal{G})$.*

*Proof.* We prove the stronger claim

$$(\star) \quad \mathrm{R}(\iota) = \mathsf{T} \iff \iota \text{ is realizable, for every } \iota \in \mathcal{I} \cup \mathcal{S},$$

where $\mathcal{I}$ and $\mathcal{S}$ are the set of possible items and slashed items (respectively) given $\boldsymbol{w}$ and $\mathcal{G}$. The theorem is the special case $\iota = (0, \mathrm{S}, N]$, since $(0, \mathrm{S}, N]$ is realizable iff $\mathrm{S} \overset{*}{\Rightarrow} \boldsymbol{w}$, iff $\boldsymbol{w} \in L(\mathcal{G})$.

**Induction measure.** For $\iota \in \mathcal{I} \cup \mathcal{S}$, define

$$d(\iota) \overset{\text{def}}{=} \begin{cases} j - i & \text{if } \iota = (i, A, j], \\ (j - i) - (q - p) & \text{if } \iota = (i, X, j]/(p, Y, q]. \end{cases}$$

Equivalently, $d(\iota)$ counts the input positions $\iota$'s outer non-terminal must derive (for a slashed item, the inner placeholder is excluded). By the validity constraints on (slashed) items, $d(\iota) \geq 1$. We prove $(\star)$ by strong induction on $d$, treating items and slashed items in parallel at each value of $d$.

**Base case** ($d(\iota) = 1$).

*Item.* For $\iota = (i, A, j]$ with $j = i+1$, the substring $\boldsymbol{w}_{(i,j]} = w_j$ has length one, and R returns the indicator $\mathbb{1}\{A \to w_j \in \mathcal{P}\}$. On the realizability side, the only way to obtain $A \overset{*}{\Rightarrow} w_j$ in CNF is via a single application of the unit rule $A \to w_j$, so realizability is equivalent to $A \to w_j \in \mathcal{P}$. Algorithm and realizability agree.

*Slashed item.* For $\iota = (i, X, j]/(p, Y, q]$ with $(j-i) - (q-p) = 1$, exactly one position of $(i, j]$ lies outside $(p, q]$. Either $p = i$ and $q = j - 1$ (the outer position is $j$) or $p = i+1$ and $q = j$ (the outer position is $i+1$). For $p = i$ and $q = j-1$, R returns:

$$\bigvee_{Z \in \mathcal{N}} X \to YZ \in \mathcal{P} \wedge Z \to w_j \in \mathcal{P} \tag{11}$$

In the symmetric case where $p = i+1$ and $q = j$, R returns:

$$\bigvee_{Z \in \mathcal{N}} X \to ZY \in \mathcal{P} \wedge Z \to w_{i+1} \in \mathcal{P} \tag{12}$$

Realizability requires $X \overset{*}{\Rightarrow} \boldsymbol{w}_{(i,p]} Y \boldsymbol{w}_{(q,j]}$. In the right-end case ($p = i$, $q = j-1$) this reduces to $X \overset{*}{\Rightarrow} Yw_j$, achievable in CNF only via $X \to YZ$ followed by $Z \to w_j$ for some $Z$ — exactly Eq. (11). The left-end case yields Eq. (12).

**Inductive step** ($d(\iota) = m \geq 2$). Assume ($\star$) holds for every $\iota'$ with $d(\iota') < m$.

*Case 1: item $\iota = (i, A, j]$ with $d(\iota) = m$.*

**Forward** ($\Rightarrow$). Suppose $R(\iota) = T$. The algorithm succeeded along one of two branches, SPLIT or GAP.

- SPLIT. Some rule $A \to BC$ and split $k \in (i, j)$ satisfy $R((i, B, k]) = R((k, C, j]) = T$. The sub-items $(i, B, k]$ and $(k, C, j]$ have as induction measure $d((i, B, k]) = k - i$ and $d((k, C, j]) = j - k$ respectively, both in $[1, m-1]$. By the induction hypothesis, $B \overset{*}{\Rightarrow} \boldsymbol{w}_{(i,k]}$ and $C \overset{*}{\Rightarrow} \boldsymbol{w}_{(k,j]}$. By the calculus of §3, $A \overset{*}{\Rightarrow} BC \overset{*}{\Rightarrow} \boldsymbol{w}_{(i,k]} \boldsymbol{w}_{(k,j]} = \boldsymbol{w}_{(i,j]}$.
- GAP. Some inner item $(k, Y, \ell]$ with $(k, \ell] \subsetneq (i, j]$ satisfies $R((i, A, j]/(k, Y, \ell]) \wedge R((k, Y, \ell]) = T$. The sub-item has $d((k, Y, \ell]) = \ell - k$, and the sub-slashed item has $d((i, A, j]/(k, Y, \ell]) = m - (\ell - k)$; strict containment of $(k, \ell]$ in $(i, j]$ forces both into $[1, m-1]$. By the induction hypothesis, $Y \overset{*}{\Rightarrow} \boldsymbol{w}_{(k,\ell]}$ and $A \overset{*}{\Rightarrow} \boldsymbol{w}_{(i,k]} Y \boldsymbol{w}_{(\ell,j]}$; substituting yields $A \overset{*}{\Rightarrow} \boldsymbol{w}_{(i,j]}$.

**Reverse** ($\Leftarrow$). Suppose $\iota$ is realizable, witnessed by a parse tree $T$. Since $m \geq 2$, the root necessarily applies some rule $A \to BC$, splitting the yield at a unique $k \in (i, j)$ into $B \overset{*}{\Rightarrow} \boldsymbol{w}_{(i,k]}$ and $C \overset{*}{\Rightarrow} \boldsymbol{w}_{(k,j]}$. The two sub-items are realizable with $d < m$; by the induction hypothesis, R returns T on each, and the SPLIT rule of $R(\iota)$ with $A \to BC$ and the split $k$ therefore returns T.

*Case 2: slashed item $\iota = (i, X, j]/(p, Y, q]$ with $d(\iota) = m$.*

**Forward** ($\Rightarrow$). Suppose $R(\iota) = T$. The algorithm succeeded along one of two branches, SPLIT or GAP.

- SPLIT. Some rule $X \to AB$ and split $k \in (i, j)/(p, q]$, with the inner $(p, q]$ on (WLOG) the $A$-side ($q < k$), satisfy $R((i, A, k]/(p, Y, q]) \wedge R((k, B, j]) = T$. The sub-slashed item has $d((i, A, k]/(p, Y, q]) = (k - i) - (q - p) < m$ (since $k < j$). The sub-item $(k, B, j]$ has $d((k, B, j]) = j - k$; an equality $j - k = m$ would force $k = q$ and $p = i$, but then $A$ would derive $\boldsymbol{w}_{(i,p]} = \varepsilon$, ruled out by CNF (which permits $\varepsilon$ only via $S \to \varepsilon$ and never on the right-hand side of a binary rule). Hence the sub-item's induction measure $d$ is also $< m$. By the induction hypothesis, $A \overset{*}{\Rightarrow} \boldsymbol{w}_{(i,p]} Y \boldsymbol{w}_{(q,k]}$ and $B \overset{*}{\Rightarrow} \boldsymbol{w}_{(k,j]}$, so via $X \to AB$, $X \overset{*}{\Rightarrow} \boldsymbol{w}_{(i,p]} Y \boldsymbol{w}_{(q,j]}$. The symmetric case (inner on $B$'s side) is analogous.
- GAP. Some item $(k, Z, \ell]$ with $(p, q] \subsetneq (k, \ell] \subsetneq (i, j]$ satisfies $R((i, X, j]/(k, Z, \ell]) \wedge R((k, Z, \ell]/(p, Y, q]) = T$. The first sub-slashed has $d((i, X, j]/(k, Z, \ell]) = m - ((\ell - k) - (q - p)) < m$ (since $\ell - k > q - p$); the second has $d((k, Z, \ell]/(p, Y, q]) = (\ell - k) - (q - p) < m$ (since $(k, \ell] \subsetneq (i, j]$ and the inner $(p, q]$ is strictly inside). By the induction hypothesis, $X \overset{*}{\Rightarrow} \boldsymbol{w}_{(i,k]} Z \boldsymbol{w}_{(\ell,j]}$ and $Z \overset{*}{\Rightarrow} \boldsymbol{w}_{(k,p]} Y \boldsymbol{w}_{(q,\ell]}$. Substituting the second into the first gives $X \overset{*}{\Rightarrow} \boldsymbol{w}_{(i,p]} Y \boldsymbol{w}_{(q,j]}$.

**Reverse** ($\Leftarrow$). Suppose $\iota$ is realizable, with parse tree $T_X$ rooted at $X$ and a distinguished leaf labeled $Y$ at yield $(p, q]$. The root of $T_X$ applies some rule $X \to AB$. The $Y$-leaf lies in $A$'s subtree or $B$'s subtree; WLOG in $A$'s, and let $k$ be the boundary between $A$'s and $B$'s yields. Then $(i, A, k]/(p, Y, q]$ is realizable (with $d = (k - i) - (q - p) < m$) and $(k, B, j]$ is realizable (with $d = j - k < m$ by the CNF argument above). By the induction hypothesis, R returns T on both, so the SPLIT rule of $R(\iota)$ with $X \to AB$ and split $k$ returns T.

**Conclusion.** By strong induction, $(\star)$ holds for every $\iota \in \mathcal{I} \cup \mathcal{S}$. Specializing to $\iota = (0, \mathrm{S}, N]$ gives $\mathrm{R}(0, \mathrm{S}, N] = \mathrm{T} \iff \boldsymbol{w} \in L(\mathcal{G})$. ∎

**Lemma 3.3** (Parallel Runtime). *Let $\boldsymbol{w} \in \Sigma^*$ be a string of length $N$ and let $\mathcal{G}$ be a context-free grammar.* $\mathrm{R}((0, \mathrm{S}, N])$ *terminates in at most* $2\lceil \log_2(2N) \rceil + \mathcal{O}(1)$ *recursive steps when* $\boldsymbol{w} \in L(\mathcal{G})$.

*Proof.* With $\boldsymbol{w} \in L(\mathcal{G})$, let $T$ with root $(0, \mathrm{S}, N]$ be a parse tree witnessing $\boldsymbol{w}$'s membership; $T$ has $M \stackrel{\text{def}}{=} 2N - 1$ items. $\mathrm{R}((0, \mathrm{S}, N])$ runs in parallel time equal to the depth of its recursion tree; thus, it suffices to exhibit, for each subproblem, a balanced decomposition the *guess* branches of $\mathrm{R}$ can choose; the depth of $\mathrm{R}$'s recursion is then bounded by the depth of this balanced decomposition.

By Lem. 3.2, $T$ has a centroid node $\iota$ whose removal partitions $T$ into components, each of size at most $\lceil M/2 \rceil$. Since $T$ is binary, $\iota$ has at most three neighbors (a parent and two children), so removing $\iota$ yields at most three components: the *parent-side* component $P$ (empty when $\iota$ is the root) and the two child subtrees $L, R$ (the subtrees rooted at $\iota$'s left and right children). In the context of a parse tree, the centroid node is either the root or an internal node. We consider both cases.

**(i) Centroid node is root.** The root node is associated with the item $(0, \mathrm{S}, N]$. Because removing the root of the binary tree $T$ yields exactly the two children's subtrees, we can use the SPLIT branch to halve the tree. Concretely, $\mathrm{R}$ breaks down $(0, \mathrm{S}, N]$ via the SPLIT rule by producing two recursive calls on items $(0, \mathrm{X}, k]$ and $(k, \mathrm{Y}, N]$ for a rule $\mathrm{S} \to \mathrm{XY} \in \mathcal{P}$ and an index $k \in (0, N)$. Then, the parse trees associated with $(0, \mathrm{X}, k]$ and $(k, \mathrm{Y}, N]$ have at most $M/2$ nodes by Lem. 3.2. This case advances by exactly one recursion level.

**(ii) Centroid node is an internal node of $T$.** Every internal node of $T$ corresponds to some item $(i, \mathrm{X}, j]$. Lem. 3.2 is a statement about an unrooted tree, so removing the centroid yields one component per neighbor of $\iota$; in our binary tree $T$ the internal node $\iota$ has at most three neighbors (its parent and its two children), so removal yields up to three components, each of size $\leq \lceil M/2 \rceil$. Once we view $T$ as rooted at $(0, \mathrm{S}, N]$, these acquire labels—the parent-side component $P$ and the two child subtrees $L, R$—but every recursive call of $\mathrm{R}$ is on a rooted item, so the algorithm can only manipulate two-piece decompositions, not three. We must therefore fuse Jordan's three pieces into two cuts the algorithm can perform. The GAP rule realizes the fusion $P$ vs. $\{\iota\} \cup L \cup R$ by cutting the edge above $\iota$, but the inner side can be up to $M - 1$ nodes (if $|P| = 1$). The SPLIT rule realizes the fusion $L$ vs. $R$ by removing $\iota$, but only upon recursing on $\iota$ itself. We therefore charge *two* recursion levels: a GAP cut to isolate $\{\iota\} \cup L \cup R$ (creating $P$ and the item at $\iota$ with $|P| + 1 \leq \lceil M/2 \rceil + 1$), then a SPLIT cut at $\iota$ to break $\iota$'s subtree into $L$ and $R$ (both with $|L|, |R| \leq \lceil M/2 \rceil$). After these two levels, the remaining subproblems are exactly Jordan's three pieces $P, L, R$, each of size at most $\lceil M/2 \rceil + 1$.

- *Level 1 (GAP on $\iota$).* $\mathrm{R}$ dispatches via the GAP rule with inner item $(i, \mathrm{X}, j]$ corresponding to the centroid node $\iota$, producing recursive calls on *(a)* the outer slashed item $(0, \mathrm{S}, N]/(i, \mathrm{X}, j]$, whose associated tree is $P$ with $\iota$'s subtree replaced by a placeholder leaf, of size $|P| + 1 \leq \lceil M/2 \rceil + 1$; and *(b)* the inner item $(i, \mathrm{X}, j]$, whose associated tree is $\{\iota\} \cup L \cup R$ of size $|L| + |R| + 1$. The outer call is already balanced by Lem. 3.2, but the inner call may still be nearly all of $T$ (when $|P|$ is small), so we charge a second level to it.

- *Level 2 (SPLIT on $\iota$).* Let $\mathrm{X} \to \mathrm{YZ}$ be the production at $\iota$ in the parse tree, and let $k$ be the split index between $\iota$'s left and right children. $\mathrm{R}$ dispatches via the SPLIT rule on $(i, \mathrm{X}, j]$, producing recursive calls on $(i, \mathrm{Y}, k]$ and $(k, \mathrm{Z}, j]$, whose associated trees are exactly $L$ and $R$, both of size at most $\lceil M/2 \rceil$ by Lem. 3.2.

After these two levels, every remaining subproblem has associated tree of size at most $\lceil M/2 \rceil + 1$.

We now iterate the above on each subproblem. Each recursive call is on either an item or a slashed item, so we must argue the centroid bound for both kinds:

- *Item subproblem.* The recursive call is on an item $(i', \mathrm{X}', j']$, whose parse tree is a subtree of $T$. The case analysis above (cases (i)–(ii)) applies verbatim.

- *Slashed-item subproblem.* The recursive call is on a slashed item $(i', \mathrm{X}', j']/(p, \mathrm{Y}, q]$. Associate to it the parse tree of $\mathrm{X}'$ over $\boldsymbol{w}_{(i', j']}$ with a subtree yielding $\boldsymbol{w}_{(p, q]}$ replaced by a placeholder leaf; call this $T'$. The internal nodes of $T'$ are again items, and $T'$ is binary, so Lem. 3.2 applies to $T'$. The case split mirrors (i)–(ii) with the slashed-item rules of Alg. 2: if the centroid is the root of $T'$, $\mathrm{R}$ takes the slashed-item SPLIT branch and advances one level; if the centroid is

an internal node of $T'$, R takes the slashed-item GAP branch (peeling off the centroid's subtree as an inner slashed item) followed by a slashed-item SPLIT on the resulting subproblem, again advancing two levels and leaving every surviving subproblem of associated tree size at most $\lceil |T'|/2 \rceil + 1$.

A *recursion level* is one step of R's call tree: every dispatch through the SPLIT or GAP branch adds exactly one level (its recursive calls live one level deeper), and the parallel depth of R equals the depth of this call tree. With this counting, we call one application of either case (i) or case (ii) a *centroid round*: case (i) is a single SPLIT dispatch and so consumes one recursion level, while case (ii) is a GAP dispatch *followed by* a SPLIT dispatch on the inner item it produced, and therefore consumes two. Define the *shape size* $s(\iota)$ of a subproblem as the number of items in its associated tree (treating a placeholder leaf as one node). Inductively, every centroid round shrinks the shape size by a factor of two up to an additive constant: a subproblem of shape size $m$ produces subproblems of shape size at most $\lceil m/2 \rceil + 1$. Lem. 3.2 bounds the surviving pieces $L, R$ by $\lceil m/2 \rceil$ directly, and the parent-side piece $P$ by $\lceil m/2 \rceil$ as well; the only loss is the placeholder leaf adjoined to $P$, contributing one extra node to that piece. Starting from $s((0, \mathrm{S}, N)) = M = 2N - 1$, the recurrence $s_{r+1} \leq \lceil s_r/2 \rceil + 1$ unrolls (geometric series) to $s_r \leq M/2^r + \mathcal{O}(1)$ for every $r \geq 0$. The base case $s(\iota) = 1$ (a length-1 item or a slashed item whose outer tree is just a placeholder leaf) is handled in R's leaf branches without further recursion, so it suffices to pick the smallest $r$ for which every surviving subproblem has constant shape size, which happens at $r = \lceil \log_2 M \rceil + \mathcal{O}(1)$. Since each round costs at most two recursion levels, R's recursion has depth at most $2\lceil \log_2 M \rceil + \mathcal{O}(1) \leq 2\lceil \log_2(2N) \rceil + \mathcal{O}(1)$, matching the bound in the theorem statement. ∎

### B.2. Encoding (Slashed) Items in Padding Symbols

Our constructions associate padding symbols with distinct objects. For example, when computing the realizability of items in Alg. 1 and Alg. 2 on AHAT$s$, we associate each padding symbol with an item. To enable this, we introduce a novel theoretical gadget implementable by AHAT$s$ that enables a padding symbol at some position $i$ to compute the encoding of its associated items from the unique position $i$:

**Lemma B.1** (Converting a padding-symbol position into an encoding of structured data). *Fix a string length $N$, and let*

$$\mathcal{A} \stackrel{\text{def}}{=} \mathcal{A}^{(1)} \times \mathcal{A}^{(2)} \times \cdots \times \mathcal{A}^{(m)}$$

*be a Cartesian product of $m$ finite sets, where each factor $\mathcal{A}^{(j)}$ is either (i) a fixed finite alphabet (e.g. $\mathcal{N}$) of size $\mathcal{O}(1)$ in $N$, or (ii) the index set $(0, N]$. The set $\mathcal{A}$ has size $|\mathcal{A}| = |\mathcal{A}^{(1)}| \times \ldots |\mathcal{A}^{(m)}|$, and we fix any enumeration of its elements*

$$\mathcal{A} = \{s_1, s_2, \ldots, s_{|\mathcal{A}|}\}, \qquad s_i = \left( s_i^{(1)}, s_i^{(2)}, \ldots, s_i^{(m)} \right) \in \mathcal{A}$$

*that arises from interpreting the $\log|\mathcal{A}| = \mathcal{O}(\log N)$ bits of $i \in \{0, 1, \ldots, |\mathcal{A}| - 1\}$ as the concatenation of $m$ field-encodings, one per factor $\mathcal{A}^{(j)}$. Then there exists an AHAT transformer $\mathsf{T}$ of constant depth, operating on an input padded with $|\mathcal{A}|$ padding symbols, such that at the residual stream of the $i$-th padding symbol, $\mathsf{T}$ stores the layer-norm hashes*

$$\phi\left(s_i^{(1)}\right), \ \phi\left(s_i^{(2)}\right), \ \ldots, \ \phi\left(s_i^{(m)}\right).$$

*Proof.* The argument has three steps: (1) make $\phi(i)$ and $\phi(N)$ available at each padding symbol; (2) extract the bits of $i$ corresponding to each factor $\mathcal{A}^{(j)}$; and (3) store the layer-norm hash of each extracted value.

**Step 1: making $\phi(i)$ and $\phi(N)$ available.** At the $i$-th padding symbol, the value $\phi(i)$ can be added to the residual stream by uniform attention over the strict left context, which counts positions; analogously, $\phi(N)$ is obtained by uniform attention restricted to the string symbols, which counts $N = |\boldsymbol{w}|$. Both are obtained with one causally masked attention layer (Merrill & Sabharwal, 2024b).

**Step 2: extracting per-factor bit ranges.** Since $|\mathcal{A}|$ is a product of $m$ factors, each of size $\mathcal{O}(1)$ or $N$, we can write

$$|\mathcal{A}| = \prod_{j=1}^{m} M_j, \qquad M_j \in \{\mathcal{O}(1), N\}.$$

Reading $i \in \{0, 1, \ldots, |\mathcal{A}| - 1\}$ in mixed radix $(M_1, M_2, \ldots, M_m)$ yields, for each $j \in \{1, \ldots, m\}$, a coordinate

$$s_i^{(j)} \stackrel{\text{def}}{=} \left\lfloor i \, / \, \prod_{k<j} M_k \right\rfloor \bmod M_j \; \in \; \mathcal{A}^{(j)}.$$

By Merrill & Sabharwal, 2024a, Lem. 1, with a constant number of attention layers and given $\phi(i)$ and $\phi(N)$ in the residual stream, integer division and modulo by either $N$ or a constant divisor can be realized at the level of layer-norm hashes. Iterating this $m$ times (once per factor) computes $\phi\!\left(s_i^{(j)}\right)$ in the stream for each $j$.

**Step 3: depth and width.** The number of layers required at Step 2 is constant in $N$ because $m$ is constant (it depends only on $\mathcal{A}$, not on $N$), and each extraction uses a constant number of applications of Merrill & Sabharwal, 2024a, Lem. 1. The residual stream stores one hash per factor, so width is $\mathcal{O}(m) = \mathcal{O}(1)$.

**Bijection with padding positions.** The mixed-radix decomposition above is a bijection between $\{0, 1, \ldots, |\mathcal{A}| - 1\}$ and $\mathcal{A}$. Hence each of the $|\mathcal{A}|$ padding symbols stores the hashes of a distinct element $s_i \in \mathcal{A}$, and every element of $\mathcal{A}$ is realized exactly once. ∎

**Examples.** Two illustrations of how the schema $\mathcal{A}$ is instantiated in the constructions:

- For items of the form $(p, \mathrm{X}, q)$, take $m = 3$ with $\mathcal{A}^{(1)} = (0, N]$, $\mathcal{A}^{(2)} = \mathcal{N}$, $\mathcal{A}^{(3)} = (0, N]$, so $|\mathcal{A}| = |\mathcal{N}| \cdot N^2$. The padding-symbol-to-item map then reads off, from $i$, the two positions $p, q$ and the non-terminal X.

- For slashed-item decompositions $(\iota, \xi) \in \mathcal{D}$, the schema is the Cartesian product of the item factors and the decomposition factors. The total size matches the $\mathcal{O}(N^6)$ bound on padding-symbol allocation in §B.3.

## B.3. General CFL Recognition on Transformers

**Theorem 3.1.** $\mathrm{CFL} \subseteq \mathrm{MAHAT}^1_6 \subseteq \mathrm{AHAT}^1_7$.

*Proof.* We give a MAHAT construction $\mathsf{T}$ that simulates R (Algs. 1 and 2) on every (slashed) item of $\boldsymbol{w}$ in parallel and reads off the bit $\mathbb{1}\{\boldsymbol{w} \in L(\mathcal{G})\}$ at EOS. The construction fixes a padding-symbol allocation, a residual-stream layout, a partition of layers into a constant-depth preamble $A$, a constant-depth looped block $B$ (looped $\mathcal{O}(\log N)$ times), and a constant-depth tail $C$ in the sense of Def. 2.3, and—for each sublayer in $A$, $B$, $C$—an explicit choice of attention head (query, key, value, mask) and feedforward map.

**Padding allocation.** Recall $\mathcal{I}$ and $\mathcal{S}$ are the items and slashed items associated with $\boldsymbol{w}$ and $\mathcal{G}$. The decomposition modes SPLIT and GAP of Algs. 1 and 2 are indexed by

$$\mathcal{D}_{\mathrm{SPLIT}} \stackrel{\text{def}}{=} \{(p, k) \mid p \in \mathcal{P}, \; k \in (0, N)\},$$
$$\mathcal{D}_{\mathrm{GAP}} \stackrel{\text{def}}{=} \{(p, \mathrm{Z}, q) \mid \mathrm{Z} \in \mathcal{N}, \; p, q \in (0, N)\},$$

and

$$\mathcal{D} \stackrel{\text{def}}{=} \{(\iota, \xi) \mid \iota \in \mathcal{I} \cup \mathcal{S}, \; \xi \in \mathcal{D}_{\mathrm{SPLIT}} \cup \mathcal{D}_{\mathrm{GAP}}\}.$$

We append $P \stackrel{\text{def}}{=} 3 \max(|\mathcal{I}|, |\mathcal{S}|, |\mathcal{D}|) = \mathcal{O}(N^6)$ padding symbols, giving an input of length $N + \mathcal{O}(N^6)$.

**Residual-stream layout.** Every position—string or padding—carries the same partitioned residual stream $\boldsymbol{x} \in \mathbb{Q}^D$ with $D = \mathcal{O}(1)$. The named slots are:

- $\boldsymbol{x}.\mathtt{kind} \in \{\mathrm{STR}, \mathrm{ITEM}, \mathrm{SLASH}, \mathrm{DEC}\}$: a constant-width one-hot identifying which family the position belongs to (string symbol, item, slashed item, or decomposition).
- $\boldsymbol{x}.\mathtt{pos} \in \mathbb{Q}^4$: layer-norm hash $\phi(i)$ of the position index, written in preamble $A$ via Lem. B.1.
- $\boldsymbol{x}.\mathtt{len} \in \mathbb{Q}^4$: layer-norm hash $\phi(N)$ of the string length, written in $A$ via Lem. B.1..
- For string positions: $\boldsymbol{x}.\mathtt{sym} \in \mathbb{Q}^{|\Sigma|}$, the one-hot $[\![w_i]\!]$ of the input symbol.
- For item positions associated with $\iota = (i, \mathrm{X}, j)$ (or one of the four bracket shapes): coordinate slots $\boldsymbol{x}.\mathtt{i} \stackrel{\text{def}}{=} \phi(i)$, $\boldsymbol{x}.\mathtt{j} \stackrel{\text{def}}{=} \phi(j)$, $\boldsymbol{x}.\mathtt{NT} \in \mathbb{Q}^{|\mathcal{N}|}$, $\boldsymbol{x}.\mathtt{shape} \in \mathbb{Q}^4$ (which of the four bracket shapes), and a key slot $\boldsymbol{x}.\mathtt{key} \stackrel{\text{def}}{=} \phi(\iota)$ that hashes the full item tuple. Slashed-item positions add two more coordinate slots $\boldsymbol{x}.\mathtt{p}$, $\boldsymbol{x}.\mathtt{q}$ and a nonterminal slot $\boldsymbol{x}.\mathtt{NT}_{\mathtt{in}}$ for the inner item, plus shape slots; all written in $A$.

- For decomposition positions $(\iota, \xi) \in \mathcal{D}$: a key slot $\boldsymbol{x}.\mathtt{key} \overset{\text{def}}{=} \phi(\iota, \xi)$, the inherited slots of $\iota$, slots encoding $\xi$ (the rule and split index for SPLIT; the inner item for GAP), and two "child key" slots $\boldsymbol{x}.\mathtt{key}_1 \overset{\text{def}}{=} \phi(\iota_1)$, $\boldsymbol{x}.\mathtt{key}_2 \overset{\text{def}}{=} \phi(\iota_2)$, where $\iota_1, \iota_2 \in \mathcal{I} \cup \mathcal{S}$ are the two sub-items produced by applying $\xi$ to $\iota$. The child keys are also computed in $A$.
- $\boldsymbol{x}.\mathtt{v} \in \{\mathrm{T}, \mathrm{F}, \bot\} \subseteq \mathbb{Q}^3$: the three-valued realizability bit for the associated (slashed) item (at item / slashed-item positions) or the per-decomposition conjunction (at decomposition positions). Initialized to $\bot$ at all padding positions in $A$.
- Scratch slots $\boldsymbol{x}.\mathtt{v}_1, \boldsymbol{x}.\mathtt{v}_2, \boldsymbol{x}.\mathtt{v}_{\mathtt{dec}} \in \mathbb{Q}^3$ used inside the loop body to stage reads; zeroed at the end of each iteration.

All hashes live in $\mathbb{Q}^4$, all one-hots are constant-dimensional, and the slots above sum to $\mathcal{O}(1)$ width, satisfying the MAHAT definition. When we say a feedforward network "reads $\mathtt{key}$ and writes $\mathtt{key}_1$," we mean the underlying linear projections target the corresponding sub-block of $\boldsymbol{x}$ using multi-pre-norm.

**Preamble $A$ (constant depth, runs once).** Block $A$ assembles every slot above and computes the base cases of R. Its sublayers, in order, are:

1. *Position hashes.* A first sublayer makes the position hashes $\mathtt{pos} \overset{\text{def}}{=} \phi(i)$ and $\mathtt{len} \overset{\text{def}}{=} \phi(N)$ available at every position: one causally-masked attention head uniformly attends to all positions to count $i$ (and hence write $\phi(i)$), and a second head uniformly attends to string positions only to count $N = |\boldsymbol{w}|$ (and write $\phi(N)$), by the layer-norm-hash construction of Merrill & Sabharwal (2024a, Lem. 1) (cf. Def. A.2).

2. *Position-dependent tagging.* We interleave the three padding families by position residue rather than laying them out in contiguous blocks. We assign the family of each padding position by residue mod 3 among padding positions:

$$\mathtt{kind} \leftarrow \begin{cases} \text{STR} & \text{if input symbol} \neq \square, \\ \text{ITEM} & \text{if } (i - N) \mod 3 = 0, \\ \text{SLASH} & \text{if } (i - N) \mod 3 = 1, \\ \text{DEC} & \text{if } (i - N) \mod 3 = 2. \end{cases}$$

Both $\phi(i)$ and $\phi(N)$ are already in the residual stream from step 1, so $\phi((i - N) \mod 3)$ follows by one log-precision subtraction and a further application of Merrill & Sabharwal (2024a, Lem. 1) (modulo by the constant 3). A position-wise MLP then reads $\phi((i - N) \mod 3)$ together with the sign of $i - N$ (a single layer-norm-hash comparison against 0) and writes $\mathtt{kind}$ via the four-entry lookup above. The whole step uses constant depth and constant width. By our choice of $P$, each residue class modulo 3 contains at least $\max(|\mathcal{I}|, |\mathcal{S}|, |\mathcal{D}|) = |\mathcal{D}|$ positions, so every family has room for its full schema.

3. *Schema decoding.* Lem. B.1 is applied to each residue class separately, corresponding to the schemata $\mathcal{I}$, $\mathcal{S}$, $\mathcal{D}$. Constant-depth attention blocks populate the per-family coordinate, nonterminal, and decomposition slots at each padding position, together with $\mathtt{key} \overset{\text{def}}{=} \phi(\iota)$ or $\phi(\iota, \xi)$.

4. *Child keys (for decompositions).* A position-wise MLP at every $(\iota, \xi) \in \mathcal{D}$ reads its already-populated $\iota$ and $\xi$ slots and writes the two child keys $\mathtt{key}_1 \overset{\text{def}}{=} \phi(\iota_1), \mathtt{key}_2 \overset{\text{def}}{=} \phi(\iota_2)$, where $\iota_1, \iota_2$ are read off $\xi$ as in Algs. 1 and 2: for SPLIT with $\xi = (\mathrm{A} \to \mathrm{BC}, k)$ and $\iota = [i, \mathrm{X}, j]$, we set $\iota_1 = [i, \mathrm{B}, k], \iota_2 = [k, \mathrm{C}, j]$ if $\mathrm{X} = \mathrm{A}$ and $(\iota_1, \iota_2) = (\iota, \iota)$ otherwise (a guard that will make $\mathtt{v} = \bot$ permanently for this decomposition); for GAP with $\xi = (p, \mathrm{Y}, q)$, we set $\iota_1 = \iota/(p, \mathrm{Y}, q]$ and $\iota_2 = (p, \mathrm{Y}, q]$. Both are finite mappings over the constant-size sets $\{\text{SPLIT}, \text{GAP}\} \times \mathcal{P} \times \mathcal{N} \times \{\text{shapes}\}$ combined with the linear arithmetic on position hashes provided by Lem. B.1, hence implementable by a constant-depth feedforward network (Yang et al., 2025).

5. *Initial-$\mathtt{v}$ writing (base cases).* A position-wise MLP can first evaluate which (slashed) item positions correspond to base cases by comparing the item indices. The MLP then writes $\mathtt{v}$ at every padding position as follows:
   - At an item position $\iota = (i - 1, \mathrm{X}, i]$ (a length-1 item), the position first attends with a causally-masked head, gated by the layer-norm-hash equality $\phi(i) \cdot \mathtt{pos} = 1$, to the unique string position holding $w_i$ and reads $[\![w_i]\!]$ into a scratch slot (Merrill & Sabharwal, 2024b). An MLP then writes $\mathtt{v} \leftarrow \mathbb{1}\{\mathrm{X} \to w_i \in \mathcal{P}\}$ via the finite map $\mathcal{N} \times \Sigma \to \{\mathrm{T}, \mathrm{F}\}$ implementable by a feedforward network (Yang et al., 2025).
   - At a slashed-item base case $\iota = (i, \mathrm{X}, j]/(i, \mathrm{Y}, j - 1]$ (or its symmetric variant), the same idea applies: read $w_j$ and write $\mathtt{v} \leftarrow \mathbb{1}\{\exists \mathrm{Z}. \mathrm{X} \to \mathrm{YZ} \in \mathcal{P} \land \mathrm{Z} \to w_j \in \mathcal{P}\}$, a finite map implementable by a feedforward network (Yang et al., 2025).
   - All other item, slashed-item, and decomposition positions get $\mathtt{v} \leftarrow \bot$.

Each step uses $\mathcal{O}(1)$ attention and feedforward sublayers, so $A$ has constant total depth. **Loop body $B$ (constant depth, looped $\mathcal{O}(\log N)$ times).** One iteration of $B$ updates $\mathtt{v}$ at every padding position by one round of three-valued R, fusing the

"decomposition resolution" and "item aggregation" phases of the algorithm. It consists of three sub-blocks; every attention head below is unmasked and uses the dot-product score with the hardmax projection of Def. A.1, gated by layer-norm-hash equality (Def. A.2).

*Sub-block $B_{\text{READ}}$ (read children's v).* At each decomposition position $(\iota, \xi)$:

- Head $H^1_{\text{READ}}$: query $\boldsymbol{q} \stackrel{\text{def}}{=} \text{key}_1$, key $\boldsymbol{k}_u \stackrel{\text{def}}{=} \boldsymbol{x}_u.\text{key}$, value $\boldsymbol{v}_u \stackrel{\text{def}}{=} \boldsymbol{x}_u.\text{v}$. The score peaks at $u = \iota_1$, so hardmax writes v of the first child into the scratch slot $\text{v}_1$.
- Head $H^2_{\text{READ}}$: identical with query $\text{key}_2$, writing the second child's v into $\text{v}_2$.

*Sub-block $B_{\text{CONJ}}$ (three-valued conjunction at decomposition positions).* We first devise in Tab. 3 a natural[7] truth table that extends standard Boolean logic with the unknown truth value $\perp$.

| $P$ | $Q$ | $P \wedge_3 Q$ | $P \vee_3 Q$ |
|---|---|---|---|
| T | $\perp$ | $\perp$ | T |
| $\perp$ | T | $\perp$ | T |
| F | $\perp$ | F | $\perp$ |
| $\perp$ | F | F | $\perp$ |
| $\perp$ | $\perp$ | $\perp$ | $\perp$ |

*Table 3.* Truth table for a three-valued logic
that handles propositions with unknown truth value.

A position-wise MLP $F_{\text{CONJ}}$ at every $(\iota, \xi)$ writes

$$\text{v}_{\text{dec}} \leftarrow \text{v}_1 \wedge_3 \text{v}_2$$

using the truth table in Tab. 3. The map $\{\text{T}, \text{F}, \perp\}^2 \to \{\text{T}, \text{F}, \perp\}$ has 9 entries, so a constant-width two-layer ReLU network suffices (Yang et al., 2025). Monotonicity is built into the table: $\text{v}_{\text{dec}}$ never moves away from $\{\text{T}, \text{F}\}$ once written.

*Sub-block $B_{\text{DISJ}}$ (three-valued disjunction over decompositions, written back to item positions).* At each item or slashed-item position $\iota$, we want

$$\text{v} \leftarrow \bigvee_3 \{\text{v}_{\text{dec}(\iota, \xi)} \mid \xi \in \mathcal{D}_{\text{SPLIT}} \cup \mathcal{D}_{\text{GAP}}\},$$

the three-valued disjunction, defined by the same table: T if any disjunct is T, F if all are F, $\perp$ otherwise. Two unmasked attention heads keyed on $\iota$'s identity realize this:

- Head $H^{\text{T}}_{\text{DISJ}}$: query $\boldsymbol{q} \stackrel{\text{def}}{=} \text{key}$ at $\iota$, key $\boldsymbol{k}_u \stackrel{\text{def}}{=} \boldsymbol{x}_u.\text{key}|_\iota$ (i.e., the embedded $\iota$-component of the decomposition's key, extracted by a fixed linear projection) at every decomposition $u = (\iota', \xi)$, value $\boldsymbol{v}_u \stackrel{\text{def}}{=} \mathbb{1}\{\boldsymbol{x}_u.\text{v}_{\text{dec}} = \text{T}\}$. The score is maximized exactly at decompositions of $\iota$, and hardmax's output is nonzero iff at least one such decomposition has $\text{v}_{\text{dec}} = \text{T}$. A subsequent MLP writes this bit into a scratch slot $a^{\text{T}}$.
- Head $H^{\perp}_{\text{DISJ}}$: identical, but with value $\boldsymbol{v}_u \stackrel{\text{def}}{=} \mathbb{1}\{\boldsymbol{x}_u.\text{v}_{\text{dec}} = \perp\}$; result written into $a^{\perp}$.

A final MLP $F_{\text{DISJ}}$ then writes

$$\text{v} \leftarrow \begin{cases} \text{T} & \text{if } \text{v} = \text{T} \text{ or } a^{\text{T}} = 1, \\ \text{F} & \text{if } \text{v} \neq \text{T} \text{ and } a^{\text{T}} = 0 \text{ and } a^{\perp} = 0, \\ \text{v} & \text{otherwise (kept at } \perp \text{ or its already-decided value).} \end{cases}$$

The first clause enforces monotonicity: once $\text{v} \in \{\text{T}, \text{F}\}$ it never changes. The scratch slots $\text{v}_1, \text{v}_2, \text{v}_{\text{dec}}, a^{\text{T}}, a^{\perp}$ are zeroed by a final position-wise linear map, so $B$ is length-preserving.

**Tail $C$ (constant depth, runs once).** A single unmasked attention head at EOS: query $\boldsymbol{q} \stackrel{\text{def}}{=} \phi((0, \text{S}, N])$ ($0$ and $\text{S}$ are constant, $N$ is computable via attention), key $\boldsymbol{k}_u \stackrel{\text{def}}{=} \boldsymbol{x}_u.\text{key}$, value $\boldsymbol{v}_u \stackrel{\text{def}}{=} \boldsymbol{x}_u.\text{v}$. The head concentrates on the unique padding position

---

[7]Intuitively, a $\vee_3$ commits to T as soon as any disjunct is T (regardless of unknowns $\perp$ elsewhere). Similarly, $\wedge_3$ commits to F as soon as any conjunct is F. The result otherwise stays $\perp$.

whose key matches, copying $v_{(0,\mathrm{S},N]}$ into the EOS residual stream. The linear classifier $c$ accepts iff this slot equals the T encoding.

**Looping schedule.** Setting $\ell_1, \ell_2$ in Def. 2.3 to bracket $A$, $B$, $C$ and looping $B$ for $T \overset{\text{def}}{=} 2\lceil \log_2(2N) \rceil + \mathcal{O}(1)$ iterations (Lem. 3.3), the transformer computes

$$
C \circ B^{\mathcal{O}(\log N)} \circ A \circ \mathsf{emb}\left( \boldsymbol{w} \underbrace{\square \cdots \square}_{\mathcal{O}(N^6)} \right).
$$

**Correctness invariant.** For $\iota \in \mathcal{I} \cup \mathcal{S}$ define the depth-$\tau$ truncation $\mathrm{R}^{(\tau)}(\iota)$ inductively: $\mathrm{R}^{(0)}(\iota)$ is $\mathrm{R}(\iota)$ if $\iota$ is a base case and $\bot$ otherwise; $\mathrm{R}^{(\tau)}(\iota)$ evaluates the body of Algs. 1 and 2 on $\iota$ with each recursive call replaced by $\mathrm{R}^{(\tau-1)}$ of the child item, with all $\wedge$ and $\vee$ interpreted via Tab. 3. Monotonic stabilization holds: $\mathrm{R}^{(\tau)}(\iota) \in \{\mathrm{T}, \mathrm{F}\} \Rightarrow \mathrm{R}^{(\tau+1)}(\iota) = \mathrm{R}^{(\tau)}(\iota)$.

We prove by induction on $\tau$ that

$$
P(\tau): \quad \text{after } \tau \text{ iterations of } B, \text{ every item/slashed-item position carries } \mathtt{v} = \mathrm{R}^{(\tau)}(\iota).
$$

*Base case $\tau = 0$.* Step (v) of $A$ writes $\mathtt{v} = \mathrm{R}^{(0)}(\iota)$ at base-case items and slashed items and $\mathtt{v} = \bot$ everywhere else, exactly matching $\mathrm{R}^{(0)}$.

*Inductive step.* Assume $P(\tau)$. At iteration $\tau + 1$, sub-block $B_{\mathrm{READ}}$ at every $(\iota, \xi)$ reads its two children's $\mathtt{v}$ values, which by $P(\tau)$ are $\mathrm{R}^{(\tau)}(\iota_1)$ and $\mathrm{R}^{(\tau)}(\iota_2)$. Sub-block $B_{\mathrm{CONJ}}$ writes $\mathtt{v}_{\mathtt{dec}(\iota,\xi)} = \mathrm{R}^{(\tau)}(\iota_1) \wedge_3 \mathrm{R}^{(\tau)}(\iota_2)$ at every decomposition. Sub-block $B_{\mathrm{DISJ}}$ then aggregates these per-decomposition conjunctions at every item position via the three-valued $\bigvee_3$, with the monotonicity guard preserving any already-decided $\mathtt{v}$. By the definition of $\mathrm{R}^{(\tau+1)}$ as the depth-$\tau$ unrolling of Algs. 1 and 2 under three-valued logic,

$$
\mathtt{v} \leftarrow \bigvee\nolimits_3 \left\{ \mathrm{R}^{(\tau)}(\iota_1) \wedge_3 \mathrm{R}^{(\tau)}(\iota_2) \mid \xi \right\} = \mathrm{R}^{(\tau+1)}(\iota),
$$

so $P(\tau + 1)$ holds. By Lemmata 3.1 and 3.3, $\boldsymbol{w} \in L(\mathcal{G})$ implies that $\mathrm{R}^{(T)}((0, \mathrm{S}, N]) = \mathrm{R}((0, \mathrm{S}, N]) = \mathrm{T}$ for depth $T = \mathcal{O}(\log N)$. Hence after the $\mathcal{O}(\log N)$ looped iterations the padding position for $(0, \mathrm{S}, N]$ stores T iff $\boldsymbol{w} \in L(\mathcal{G})$ (and otherwise stores F or $\bot$), which $C$ copies to EOS and $c$ reads off.

**Resource accounting.**

- *Padding:* $|\mathcal{I}| = \mathcal{O}(N^2)$ item symbols, $|\mathcal{S}| = \mathcal{O}(N^4)$ slashed-item symbols, $|\mathcal{D}| = \mathcal{O}(N^6)$ decomposition symbols, leading to $P = \mathcal{O}(N^6)$.
- *Depth:* $A$ is $\mathcal{O}(1)$ sublayers, $B$ is $\mathcal{O}(1)$ sublayers, $C$ is one sublayer, and $B$ is looped $\mathcal{O}(\log N)$ times, giving $\mathcal{O}(\log N)$ total depth.
- *Heads:* causally-masked for padding-to-string reads (position hashes, base-case symbol look-up); unmasked for inter-padding attention (child reads in $B_{\mathrm{READ}}$, disjunction in $B_{\mathrm{DISJ}}$, accept head in $C$). This places $\mathsf{T}$ in $\mathrm{MAHAT}_6^1$.

Applying Lem. 2.1 yields $\mathrm{CFL} \subseteq \mathrm{MAHAT}_6^1 \subseteq \mathrm{AHAT}_7^1$. ∎

## B.4. Unambiguous CFL Recognition on Transformers

We first define two pieces of notation. A variable-free Boolean formula $\psi$ in **postfix notation** is a string over the alphabet $\{\mathrm{T}, \mathrm{F}, \neg, \wedge, \vee\}$, defined recursively: T and F are postfix formulas, and if $\alpha, \beta$ are postfix formulas then so are $\alpha\neg$, $\alpha\beta\wedge$, and $\alpha\beta\vee$. The **binary expression tree** of $\psi$, denoted $\mathcal{G}_\psi = (\mathcal{V}_\psi, \mathcal{E}_\psi)$, is defined in lockstep with postfix: an atomic formula $w \in \{\mathrm{T}, \mathrm{F}\}$ has the single-node tree whose only vertex is a leaf labeled $w$; a unary compound $\alpha\neg$ has the tree whose root is a fresh internal node labeled $\neg$ with $\mathcal{G}_\alpha$ as its only subtree; and a binary compound $\alpha\beta\circ$ with $\circ \in \{\wedge, \vee\}$ has the tree whose root is a fresh internal node labeled $\circ$ with left subtree $\mathcal{G}_\alpha$ and right subtree $\mathcal{G}_\beta$. The $N$ inputs of $\psi$ are the leaves of $\mathcal{G}_\psi$; the internal nodes are labeled by connectives. Both $|\mathcal{V}_\psi|$ and the input length are $\mathcal{O}(N)$. Postfix order is a bijection between the symbols of $\psi$ and the nodes of $\mathcal{G}_\psi$: the $k$-th symbol of $\psi$ corresponds to the $k$-th node visited by a post-order traversal of $\mathcal{G}_\psi$.

We prove Lem. 4.1 (and as a byproduct, Thm. 4.1) in two stages, corresponding to the two lemmas below. First, Lem. B.2 shows that a fixed-depth transformer can *preprocess* a postfix input: it validates well-formedness and, at every operator position, populates pointers $\mathtt{arg_L}$, $\mathtt{arg_R}$ to that operator's operand positions in the input string. This turns the flat token sequence into a residual-stream encoding of the binary expression tree $\mathcal{G}_\psi$, with each node $v \in \mathcal{V}_\psi$ sitting at its postfix-order

position and storing its children's positions as pointers. Second, Lem. B.3 shows that, given any binary tree presented in this slotted form—type, $\texttt{arg}_L$, $\texttt{arg}_R$ at internal nodes; leaf value at leaves—an $\mathcal{O}(\log N)$-looped transformer evaluates the formula by simulating Rytter's (1985) parallel pebble game on the tree. Composing the two yields Lem. 4.1: the preprocessing of Lem. B.2 produces exactly the input Lem. B.3 requires, and the root of $\mathcal{G}_\psi$—which by postfix order is the last input position—then carries the value of $\psi$ after $\mathcal{O}(\log N)$ looped layers.

**Lemma B.2.** *Let $\psi$ be a string over $\{\mathrm{T}, \mathrm{F}, \neg, \wedge, \vee\}$ of length $N$. There exists a fixed-depth transformer in $\mathrm{AHAT}_0^0$ that (i) decides whether $\psi$ is a well-formed postfix formula, and (ii) if so, populates the residual stream at every operator position $v \in \mathcal{V}_\psi$ with pointers $\texttt{arg}_L, \texttt{arg}_R$ to its operand positions ($\texttt{arg}_L$ only for the unary case $\neg$).*

*Proof.* We write a **C-RASP** program (Yang & Chiang, 2024) that decides well-formedness and computes a binary predicate $\text{ARGUMENT}(k, i)$ which holds iff position $k$ holds an operand of the operator at position $i$. C-RASP is a programming language equivalent to **temporal counting logic** (Yang & Chiang, 2024), which is a lower bound on the expressive power of fixed-depth AHATs (Barcelo et al., 2024); thus, C-RASP $\subseteq \mathrm{AHAT}_0^0$.

A C-RASP program consists of a finite sequence of C-RASP operations $P_1, P_2, \ldots, P_l$. To signal acceptance, the last operation $P_l$ is evaluated at the last string position. Atomic C-RASP operations are $\pi_a(i)$, where $\pi_a(i) = \mathrm{T}$ iff $w_i = a$. Inductive C-RASP operations include the standard Boolean connectives and counting operations (e.g., counting the number of past positions such that a formula is satisfied and comparing integers).

We first write a C-RASP program that determines if an input formula is well-formed. A formula in postfix notation is well-formed if operators never try to consume more operands than are available and the formula ends with all operands being consumed. We can express the aforementioned procedure with the following C-RASP program.

$$\text{DEPTH}(i) = \#j \leq i[\pi_{\mathrm{T}}(j) \vee \pi_{\mathrm{F}}(j)] - \#j \leq i[\pi_\vee(j) \vee \pi_\wedge(j)] \tag{13a}$$

$$\text{WELL-FORMED}(i) = [\#j \leq i[\text{DEPTH}(j) < 1] = 0] \wedge [\text{DEPTH}(i) = 1] \tag{13b}$$

To determine if a formula is well-formed, WELL-FORMED is evaluated at the last position of the input string.

We now devise a binary predicate $\text{ARGUMENT}(k, i)$ such that $\text{ARGUMENT}(k, i) = \mathrm{T}$ iff there is an operator at position $i$ and an input argument at position $k$.

$$\text{BINARY-OP}(i) = [\pi_\wedge(i) \vee \pi_\vee(i)] \tag{14a}$$

$$\text{UNARY-OP}(i) = \pi_\neg(i) \tag{14b}$$

$$\text{DEPTH}(i) = \#j \leq i[\pi_{\mathrm{T}}(j) \vee \pi_{\mathrm{F}}(j)] - \#j \leq i[\pi_\vee(j) \vee \pi_\wedge(j)] \tag{14c}$$

$$\text{DINDEX}(i) = \#j \leq i[\text{DEPTH}(j) = \text{DEPTH}(i)] \tag{14d}$$

$$\text{PREVIOUS}(k, i) = [k = [[\#j \leq i\,\mathrm{T}] - 1]] = [k = i - 1] \tag{14e}$$

$$\text{ARGUMENT}(k, i) = [\text{UNARY-OP}(i) \wedge \text{PREVIOUS}(k, i)] \tag{14f}$$

$$\vee\,[\text{BINARY-OP}(i) \wedge [\text{PREVIOUS}(k, i) \tag{14g}$$

$$\vee\,[\text{DEPTH}(k) = \text{DEPTH}(i) \wedge [\text{DINDEX}(k) = \text{DINDEX}(i) - 1]]] \tag{14h}$$

Since C-RASP $\subseteq \mathrm{AHAT}_0^0$, this places the construction in $\mathrm{AHAT}_0^0$ with $\mathcal{O}(1)$ depth. Each operator position then uses one attention layer with ARGUMENT as its score to copy the contents of its two operand positions, populating $\texttt{arg}_L, \texttt{arg}_R$ (for the unary case $\neg$, only one operand exists, so only $\texttt{arg}_L$ is populated). ∎

**Lemma B.3.** *Let $\mathcal{G} = (\mathcal{V}, \mathcal{E})$ be a binary tree on $\mathcal{O}(N)$ nodes whose internal nodes are labeled by $\wedge$ or $\vee$ and whose leaves are labeled by $\mathrm{T}$ or $\mathrm{F}$. Suppose each node $v \in \mathcal{V}$ is assigned a distinct string position whose residual stream stores: a slot $v.\texttt{value} \in \{\mathrm{T}, \mathrm{F}, \bot\}$ initialized to the leaf value at leaves and to $\bot$ at internal nodes; $v.\texttt{type}$ holding the connective at internal nodes; and pointers $v.\texttt{arg}_L, v.\texttt{arg}_R$ to $v$'s two children at internal nodes. Then there exists a $\mathcal{O}(\log(N))$-looped transformer that, in $\mathcal{O}(\log(N))$ steps, fills $v.\texttt{value}$ at every internal node $v$ with the value of the subformula rooted at $v$.*

*Proof.* We give an explicit MAHAT construction that simulates Rytter's (1985) parallel pebble game on $\mathcal{G}$. The construction fixes a residual-stream layout, a partition of layers into a constant-depth preamble $A$, a constant-depth looped block $B$, and a constant-depth tail $C$ (cf. Def. 2.3), and—for each sublayer in $A$ and $B$—an explicit choice of attention head (query, key, value, mask) and feedforward map.

**Residual-stream layout.** Let $V \stackrel{\text{def}}{=} |\mathcal{V}| = \mathcal{O}(N)$. At every position $v \in \mathcal{V}$ the residual stream $\boldsymbol{x}_v \in \mathbb{Q}^D$ is partitioned into the following named slots (each a fixed-width sub-block that distinct projections can read and write independently with multi-pre-norm):

- $\boldsymbol{x}_v.\texttt{kind} \in \{\text{LEAF}, \text{OP}\} \subseteq \mathbb{Q}^2$: a two-bit indicator of whether $v$ is a leaf or an internal (operator) node.
- $\boldsymbol{x}_v.\texttt{type} \in \{\wedge, \vee\} \subseteq \mathbb{Q}^2$: two-bit one-hot for the connective; defined at internal nodes, zero at leaves.
- $\boldsymbol{x}_v.\texttt{value} \in \{\text{T}, \text{F}, \bot\} \subseteq \mathbb{Q}^3$: three-bit one-hot for the current value; initialized to the input value at leaves and to $\bot$ at internal nodes.
- $\boldsymbol{x}_v.\texttt{pos} \in \mathbb{Q}^4$: the layer-norm hash $\phi(v)$ of $v$'s integer position, made available by Lem. B.1; serves as the key for all equality-based attention.
- $\boldsymbol{x}_v.\texttt{arg}_{\text{L}}, \boldsymbol{x}_v.\texttt{arg}_{\text{R}} \in \mathbb{Q}^4$: the layer-norm hashes $\phi(v_1), \phi(v_2)$ of $v$'s two children's positions, given by the hypothesis (and produced by Lem. B.2); zero at leaves.
- $\boldsymbol{x}_v.\texttt{dep} \in \mathbb{Q}^4$: the layer-norm hash of $v.\texttt{dep}$'s position; initialized to $\phi(v)$ (so $v.\texttt{dep} = v$).
- $\boldsymbol{x}_v.\texttt{prop} \in \mathbb{Q}^4$: one-hot over the four possible propagators $\{\text{ID}, \neg, \mathbf{T}, \mathbf{F}\}$ (identity, negation, constant-true, constant-false); initialized to ID.
- $\boldsymbol{x}_v.\texttt{value}_{\text{L}}, \boldsymbol{x}_v.\texttt{value}_{\text{R}}, \boldsymbol{x}_v.\texttt{dep.value}, \boldsymbol{x}_v.\texttt{dep.dep}, \boldsymbol{x}_v.\texttt{dep.prop} \in \mathbb{Q}^3$ or $\mathbb{Q}^4$: scratch slots used to stage reads from neighbors within a single block; overwritten on each loop iteration.

The total width is constant, so $D = \mathcal{O}(1)$.

**Preamble $A$ (constant depth).** Block $A$ runs once before the loop and sets up the layout above. Two sublayers suffice:

1. One unmasked attention head followed by an MLP produces $\boldsymbol{x}_v.\texttt{pos} \stackrel{\text{def}}{=} \phi(v)$ at every position (cf. Lem. B.1).
2. A position-wise MLP reads the input token at $v$ and writes (i) kind, (ii) type (zero at leaves), (iii) value (the leaf value at leaves; $\bot$ at internal nodes), (iv) dep $\leftarrow$ pos, (v) prop $\leftarrow$ ID.

The hypothesis already supplies $\texttt{arg}_{\text{L}}, \texttt{arg}_{\text{R}}$.

**Loop body $B$ (constant depth, repeated $\mathcal{O}(\log N)$ times).** Each iteration of $B$ consists of three sub-blocks $B_{\text{activate}}$, $B_{\text{square}}$, $B_{\text{pebble}}$, each a constant number of attention and MLP sublayers.

*Sub-block $B_{\text{activate}}$.* Two attention heads in parallel:

- Head $H_{\text{activate}}^L$: query $\boldsymbol{q}_v \stackrel{\text{def}}{=} \boldsymbol{x}_v.\texttt{arg}_{\text{L}}$; key $\boldsymbol{k}_u \stackrel{\text{def}}{=} \boldsymbol{x}_u.\texttt{pos}$; value $\boldsymbol{v}_u \stackrel{\text{def}}{=} \boldsymbol{x}_u.\texttt{value}$. By Def. A.2, the score $\langle \boldsymbol{q}_v, \boldsymbol{k}_u \rangle$ is maximized exactly at $u = v_1$, so hardmax concentrates on $u = v_1$ and the head writes $v_1.\texttt{value}$ into $\boldsymbol{x}_v.\texttt{value}_{\text{L}}$.
- Head $H_{\text{activate}}^R$: identical but with query $\boldsymbol{x}_v.\texttt{arg}_{\text{R}}$, writing $v_2.\texttt{value}$ into $\boldsymbol{x}_v.\texttt{value}_{\text{R}}$.

A position-wise MLP $F_{\text{activate}}$ then reads $(\texttt{type}, \texttt{value}_{\text{L}}, \texttt{value}_{\text{R}}, \texttt{arg}_{\text{L}}, \texttt{arg}_{\text{R}})$ at $v$ and updates $(\texttt{dep}, \texttt{prop})$ as follows. Of the $2 \times 3 \times 3 = 18$ joint values of $(\texttt{type}, \texttt{value}_{\text{L}}, \texttt{value}_{\text{R}})$, the cases split into:

- *Exactly one of* $\texttt{value}_{\text{L}}, \texttt{value}_{\text{R}}$ *is in* $\{\text{T}, \text{F}\}$. Write $\texttt{dep} \leftarrow \texttt{arg}_{\text{R}}$ (resp. $\texttt{arg}_{\text{L}}$) and set prop from $(\texttt{type}, \texttt{value}_{\text{L}})$ (resp. $(\texttt{type}, \texttt{value}_{\text{R}})$) according to Tab. 4.
- *Both* $\texttt{value}_{\text{L}}, \texttt{value}_{\text{R}}$ *are still* $\bot$. Leave dep and prop unchanged.
- *Both are known.* Set prop to the constant $\texttt{type}(\texttt{value}_{\text{L}}, \texttt{value}_{\text{R}})$ and $\texttt{dep} \leftarrow \texttt{pos}$, so the upcoming pebble will write value directly.

| $v$'s connective | known operand value | propagator $v.\texttt{prop}$ |
|:---:|:---:|:---:|
| $\vee$ | T | $v.\texttt{prop} \equiv \text{T}$ (constant) |
| $\vee$ | F | $v.\texttt{prop}(x) = x$ (identity) |
| $\wedge$ | T | $v.\texttt{prop}(x) = x$ (identity) |
| $\wedge$ | F | $v.\texttt{prop} \equiv \text{F}$ (constant) |

*Table 4.* Defining $v$'s propagator from $v$'s connective and the known operand value, used in sub-block $B_{\text{activate}}$.

All 18 cases are a function of a constant number of bits, so $F_{\text{activate}}$ is implemented by an $\mathcal{O}(1)$-width two-layer ReLU network that realizes a truth table (Yang et al., 2025).

*Sub-block $B_{\text{square}}$.* Two attention heads in parallel:

- Head $H^1_{\text{square}}$: query $\boldsymbol{x}_v.\texttt{dep}$, key $\boldsymbol{x}_u.\texttt{pos}$, value $\boldsymbol{x}_u.\texttt{dep}$. Concentrates on $u = v.\texttt{dep}$ and writes $v.\texttt{dep.dep}$ into $\boldsymbol{x}_v.\texttt{dep.dep}$.
- Head $H^2_{\text{square}}$: same query/key, value $\boldsymbol{x}_u.\texttt{prop}$. Concentrates on $u = v.\texttt{dep}$ and writes $v.\texttt{dep.prop}$ into $\boldsymbol{x}_v.\texttt{dep.prop}$.

A position-wise feedforward network $F_{\text{square}}$ then sets $\texttt{dep} \leftarrow \texttt{dep.dep}$ and $\texttt{prop} \leftarrow \texttt{prop} \circ \texttt{dep.prop}$, where composition is the explicit $4 \times 4$ lookup table over $\{\text{ID}, \neg, \mathbf{T}, \mathbf{F}\}$; again a constant-size ReLU network suffices.

*Sub-block $B_{\textit{pebble}}$.* One attention head: query $\boldsymbol{x}_v.\texttt{dep}$, key $\boldsymbol{x}_u.\texttt{pos}$, value $\boldsymbol{x}_u.\texttt{value}$; writes $v.\texttt{dep.value}$ into $\boldsymbol{x}_v.\texttt{dep.value}$. A position-wise feedforward network $F_{\text{pebble}}$ then sets

$$\texttt{value} \leftarrow \begin{cases} \texttt{prop(dep.value)} & \text{if } \texttt{dep.value} \neq \perp, \\ \texttt{value} & \text{otherwise,} \end{cases}$$

realized again as a constant-size truth-table ReLU network over $(\texttt{prop}, \texttt{dep.value}, \texttt{value}) \in \{\text{ID}, \neg, \mathbf{T}, \mathbf{F}\} \times \{\text{T}, \text{F}, \perp\} \times \{\text{T}, \text{F}, \perp\}$.

Scratch slots $(\texttt{value}_\text{L}, \texttt{value}_\text{R}, \texttt{dep.value}, \texttt{dep.dep}, \texttt{dep.prop})$ are zeroed at the end of each iteration by an additional position-wise linear map, so block $B$ is length-preserving and self-contained.

**Tail $C$ (constant depth).** Every internal node carries its subformula's value in its $\texttt{value}$ slot. To read the truth value of the entire formula, the tail $C$ implements a single position-wise linear read-out: at the root $r$ of $\mathcal{G}$ (the last input position), the classifier $c$ inspects $\boldsymbol{x}_r.\texttt{value}$ and outputs $\mathbb{1}\{\boldsymbol{x}_r.\texttt{value} = \text{T}\}$.

**Looping schedule.** Setting $\ell_1, \ell_2$ in Def. 2.3 to bracket $A$, $B$, $C$ and choosing the loop count of $B$ to be $\lceil \log_2(V) \rceil + 1 = \mathcal{O}(\log N)$, the transformer computes
$$C \circ B^{\mathcal{O}(\log N)} \circ A \circ \texttt{emb}(\mathcal{G}).$$

**Correctness invariant.** Let $t \geq 0$ index loop iterations of $B$. Write $\texttt{dep}_v^{(t)}, \texttt{prop}_v^{(t)}, \texttt{value}_v^{(t)}$ for the contents of those slots at the end of iteration $t$ (with $t = 0$ being the state after $A$). By induction on $t$, the construction maintains Rytter's (1985) invariant

(†)   for every internal node $v$ whose subtree has depth at most $2^t$, $\texttt{value}_v^{(t)}$ equals the subformula's value.

The base case $t = 0$ is immediate as the depth-1 subtrees are exactly the leaves, whose $\texttt{value}$ is set by $A$. For the inductive step, $B_{\text{activate}}$ at iteration $t + 1$ refreshes $\texttt{dep}, \texttt{prop}$ at every still-unresolved internal node from its children's $\texttt{value}^{(t)}$ values, $B_{\text{square}}$ composes the dependency pointer and propagator with those at $\texttt{dep}$, doubling the reach, and $B_{\text{pebble}}$ writes $\texttt{value}^{(t+1)}$ whenever a value has become known. By the analysis of Rytter (1985), the reach of $\texttt{dep}$ doubles per iteration, hence (†) holds at $t + 1$.

Since $\mathcal{G}$ has depth at most $V - 1 = \mathcal{O}(N)$, after $T \stackrel{\text{def}}{=} \lceil \log_2 V \rceil + 1$ iterations (†) covers every internal node, and in particular $\texttt{value}_r^{(T)}$ at the root carries $\mathcal{G}$'s value. The total depth is $\mathcal{O}(\log N)$, placing the construction in $\text{MAHAT}_0^1$. ∎

**Lemma 4.1.** *For any variable-free postfix Boolean formula $\psi$ on $N$ inputs, there exists a $\mathcal{O}(\log(N))$-looped unpadded transformer that computes $\psi$'s value.*

*Proof.* Any boolean formula can be written in postfix notation; we assume the transformer receives the formula in this convention. The result immediately follows by composing Lem. B.2 with Lem. B.3. Concretely, the postfix input fixes a binary expression tree $\mathcal{G}_\psi = (\mathcal{V}_\psi, \mathcal{E}_\psi)$ on $\mathcal{O}(N)$ nodes, with postfix order giving a bijection between the input symbols and $\mathcal{V}_\psi$. Lem. B.2 populates each operator position's slots $\texttt{type}, \texttt{arg}_\text{L}, \texttt{arg}_\text{R}$ in $\mathcal{O}(1)$ layers; leaf positions hold their input value in $\texttt{value}$ from the start. Lem. B.3 then fills $\texttt{value}$ at every internal node of $\mathcal{G}_\psi$ in $\mathcal{O}(\log(N))$ further looped layers, with the root's value appearing at the last position of the input string (which, by postfix order, is the root of $\mathcal{G}_\psi$). ∎

**Corollary 4.1.** $\text{BFVP} \in \text{AHAT}_0^1$.

*Proof.* Immediate from Lem. 4.1: the well-formedness check provided by Lem. B.2 (invoked internally by Lem. 4.1, see §B.4) rejects ill-formed inputs, and on well-formed inputs the pebble game pebbles up $\psi$'s value in $\mathcal{O}(\log(N))$ steps, which the transformer accepts iff this value is T. Importantly, the construction does not require unmasked layers: C-RASP programs can be implemented on causally-masked layers (Yang & Chiang, 2024), while postfix notation ensures the operands of an operator always belong in its strict left context. ∎

**Theorem 4.1.** $\text{UCFL} \subseteq \text{MAHAT}_3^2 \subseteq \text{AHAT}_4^2$.

*Proof.* **Roadmap.** The proof implements the recognition algorithm of §4.1 on an MAHAT. Each *outer* iteration $t = 1, \ldots, \mathcal{O}(\log N)$ updates the marked set $\mathcal{I}_{t-1} \to \mathcal{I}_t$ by realizing Eqs. (3) and (4) as transformer layers. Concretely, one outer iteration runs four blocks:

*(1)* Build the dependency-graph edges $\mathcal{E}_t$ from the current marked set $\mathcal{I}_{t-1}$, per Eq. (3); $\mathcal{O}(1)$ layers.

*(2)* Binarize the resulting graph via the graph transform $\mathcal{T}$; $\mathcal{O}(1)$ layers.

*(3)* Evaluate, for every item $\iota$ in parallel, the Boolean disjunction $\bigvee_{\iota' \in \mathtt{R}_t(\iota)} \mathbb{1}\{\iota' \in \mathcal{I}_{t-1}\}$ of Eq. (6) by running the pebble game of Lem. B.3 on the binarized graph; $\mathcal{O}(\log N)$ layers.

*(4)* Write the resulting bit into each item's realizability slot, implementing Eq. (4); $\mathcal{O}(1)$ layers.

By Chytil et al. (1991), $\mathcal{O}(\log N)$ outer iterations suffice for $\mathcal{I}_t$ to reach the fixpoint $\mathcal{I}_*$ of §4.1, so, together with Lem. B.3, the total depth is $\mathcal{O}(\log N) \cdot \mathcal{O}(\log N) = \mathcal{O}(\log^2(N))$. The padding cost is $\mathcal{O}(N^3)$, dominated by the per-edge and per-binarization-intermediary symbols described below. After the loop a final ACCEPT block reads the realizability bit of $(0, \mathrm{S}, N]$ which equals $\mathbb{1}\{\boldsymbol{w} \in L(\mathcal{G})\}$.

Each item $(i, \mathrm{A}, j]$ is associated with a padding symbol; there are $\mathcal{O}(|\mathcal{N}|\, N^2) = \mathcal{O}(N^2)$ such item-paddings. Eq. (3) adds two symmetric families of edges. For the *right-witness* disjunct $\mathcal{R}(\iota_1, \iota_2, \iota_3)$, the still-unknown sibling shares the parent's left endpoint, so we allocate one padding symbol per pair $(i, \mathrm{A}, j], (i, \mathrm{B}, k]$ with $k \in \{i+1, \ldots, j-1\}$ and $\mathrm{B} \in \mathcal{N}$; for the *left-witness* disjunct $\mathcal{R}(\iota_2, \iota_1, \iota_3)$ the still-unknown sibling shares the parent's right endpoint, so we additionally allocate one padding symbol per pair $(i, \mathrm{A}, j], (k, \mathrm{C}, j]$ with $k \in \{i+1, \ldots, j-1\}$ and $\mathrm{C} \in \mathcal{N}$. Each family contributes $\mathcal{O}(|\mathcal{N}|\, N) = \mathcal{O}(N)$ edge-paddings per item, hence $\mathcal{O}(N^3)$ edge-paddings in total. Combining the two, the construction uses $\mathcal{O}(N^3)$ padding symbols. We leverage Lem. B.1 to enable padding symbols to add to their residual stream the encodings of their associated items from $\phi(i)$, the layer-norm hash of their position $i$.

Each item-padding allocates space to store an element in $\{\mathrm{T}, \mathrm{F}, \perp\}$ to denote that the associated item is either realizable (T), non-realizable (F) or not known yet to be realizable ($\perp$). This slot is what tracks membership in $\mathcal{I}_t$ across the outer iterations of §4.1: after block (4) of outer iteration $t$ it holds $\mathbb{1}\{\iota \in \mathcal{I}_t\}$ for the associated item $\iota$. Initially (before block (1) of $t = 1$) every padding symbol stores $\perp$.

**Initial items.** This block establishes $\mathcal{I}_0$ from §4.1 (the length-1 items witnessed by unit productions). It runs once, before the outer loop starts. An item-padding can check whether its associated item is of the form $(i - 1, \mathrm{A}, i]$ (a length-1 item, cf. Eq. (3)'s initial set $\mathcal{I}_0$) via a feedforward network that checks that the right index is one greater than the left. For all such padding symbols, another feedforward network flags whether $\mathrm{A} \to w_i \in \mathcal{P}$ to signal the realizability of that item. We can perform this procedure exactly as in the base case of §B.3. At the end of this block, each padding symbol whose associated item is in $\mathcal{I}_0$ stores $\mathrm{T}$.

**Creating the dependency graph.** This block implements step (1) of the roadmap, i.e. Eq. (3). At the start of the outer iteration $t$, the block constructs the edge set $\mathcal{E}_t$ from the realizability bits $\{\mathbb{1}\{\iota \in \mathcal{I}_{t-1}\}\}_\iota$ left in the item-paddings by the previous iteration (or by the initial-items block when $t = 1$). We instantiate Eq. (3)'s right-witness disjunct $\mathcal{R}(\iota_1, \iota_2, \iota_3)$ by identifying $\iota_3 \stackrel{\text{def}}{=} (i, \mathrm{A}, j]$ with the parent we are trying to certify, $\iota_1 \stackrel{\text{def}}{=} (i, \mathrm{B}, k]$ with the still-unknown left sibling (which will be the edge target), and $\iota_2 \stackrel{\text{def}}{=} (k, \mathrm{C}, j]$ with the right witness (which must already lie in $\mathcal{I}_{t-1}$); the combination $\mathcal{R}(\iota_1, \iota_2, \iota_3)$ then corresponds to a grammar rule $\mathrm{A} \to \mathrm{BC} \in \mathcal{P}$. Accordingly, we allocate one padding symbol per edge $\iota_3 \to \iota_1$ with $\iota_3 = (i, \mathrm{A}, j]$ and $\iota_1 = (i, \mathrm{B}, k]$; its job is to install the directed edge $\iota_3 \to \iota_1$ in $\mathcal{E}_t$ exactly when some witness $\iota_2 = (k, \mathrm{C}, j]$ certifies the combination. Since the endpoints $i, j$ and the split point $k$ are fixed by $\iota_3 \to \iota_1$, the set of candidate witnesses $\{(k, \mathrm{C}, j] \mid \mathrm{A} \to \mathrm{BC} \in \mathcal{P}\}$ has size $\mathcal{O}(|\mathcal{N}|)$ (one per nonterminal C for which the rule exists) and can be enumerated by a feedforward network from $\iota_3$ and $\iota_1$ alone. The padding symbol then performs two checks in parallel: *(i)* an attention layer that retrieves the realizability bit $\mathbb{1}\{(k, \mathrm{C}, j] \in \mathcal{I}_{t-1}\}$ from each candidate witness's item-padding, and *(ii)* a feedforward network that gates each candidate by $\mathbb{1}\{\mathrm{A} \to \mathrm{BC} \in \mathcal{P}\}$. If some candidate satisfies both, the padding symbol signals the edge $\iota_3 \to \iota_1$ in the dependency graph. Under this convention, the out-neighbors of $\iota_3$ in $(\mathcal{I}, \mathcal{E}_t)$ are exactly the items $\iota_1$ whose realizability would license $\iota_3$, so reachability from $\iota_3$ in this graph reproduces $\mathtt{R}_t(\iota_3)$ of Eq. (4). The left-witness disjunct $\mathcal{R}(\iota_2, \iota_1, \iota_3)$ of Eq. (3) is implemented symmetrically. Both edge families are constructed in parallel in the same constant-depth block.

**Binarization.** This block implements step (2) of the roadmap: given the edge set $\mathcal{E}_t$ from the previous block, it produces a binary graph $\mathcal{T}(\mathcal{I}, \mathcal{E}_t)$ whose reachability structure agrees with $(\mathcal{I}, \mathcal{E}_t)$, so that the unrolled formula $\bigvee_{v' \in \mathtt{R}_t(v)} \mathbb{1}\{\iota' \in \mathcal{I}_{t-1}\}$

of Eq. (6) can be evaluated by the pebble game of Lem. B.3. To efficiently perform reachability queries on a dependency graph, we require it to be *binary* (Rytter, 1985). To this extent, we define the *graph transform* $\mathcal{T}: \mathcal{G} \to \mathcal{G}'$ which binarizes a given directed graph $\mathcal{G}$ by adding more edges and nodes. Denoting $\mathcal{G} = (\mathcal{V}, \mathcal{E})$ and $\mathcal{G}' = (\mathcal{V}', \mathcal{E}') \stackrel{\text{def}}{=} \mathcal{T}(\mathcal{G})$, our graph transform satisfies $\mathcal{V} \subset \mathcal{V}'$ as it simply adds more nodes to the original graph. We now describe $\mathcal{T}$.

Fix a node $\iota \in \mathcal{V}$ with $M$ out-neighbors $\iota_1, \iota_2, \ldots, \iota_M \in \mathcal{V}$ in $\mathcal{G}$. If $M \leq 2$, then $\iota$ already has fan-out at most 2 and $\mathcal{T}$ leaves it untouched. Otherwise, $\mathcal{T}$ introduces $M - 2$ fresh *intermediary* nodes $h_1, h_2, \ldots, h_{M-2}$ and replaces the $M$-fold fan-out from $\iota$ by a right-deep chain of binary fan-outs: $\iota$ gets out-edges to $\iota_1$ and $h_1$; for $1 \leq i \leq M - 3$, each $h_i$ gets out-edges to $\iota_{i+1}$ and $h_{i+1}$; and the last intermediary $h_{M-2}$ gets out-edges to $\iota_{M-1}$ and $\iota_M$. Every node in this gadget has out-degree at most 2, so $\mathcal{G}'$ is binary. Reachability is preserved: the original edge $\iota \to \iota_m$ in $\mathcal{G}$ corresponds in $\mathcal{G}'$ to the directed path $\iota \to h_1 \to h_2 \to \cdots \to h_{m-1} \to \iota_m$ (for $m \geq 2$), while the edge $\iota \to \iota_1$ is preserved verbatim; so any path $\iota \to \iota_m \to \cdots$ in $\mathcal{G}$ lifts to a path $\iota \to \cdots \to \iota_m \to \cdots$ in $\mathcal{G}'$. Fig. 2 illustrates the gadget for $M = 5$.

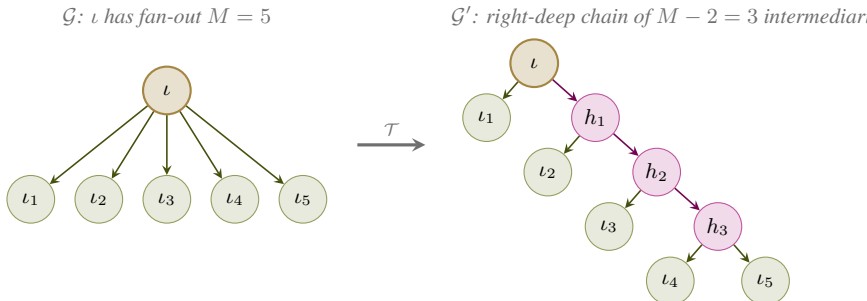

**Figure 2.** The binarization gadget $\mathcal{T}$ at a node $\iota$ with $M = 5$ out-neighbors $\iota_1, \ldots, \iota_5$. Left: the original star fan-out in $\mathcal{G}$. Right: $\mathcal{T}$ inserts $M - 2 = 3$ intermediary nodes $h_1, h_2, h_3$ to form a right-deep chain of binary fan-outs. Every node in the gadget has out-degree at most 2; the original edge $\iota \to \iota_m$ becomes the path $\iota \to h_1 \to \cdots \to h_{m-1} \to \iota_m$, preserving reachability.

The transformer construction relies on two ingredients already in place: (i) the per-edge padding symbols allocated in block (1), one per candidate directed edge $\iota_3 \to \iota_1$, each carrying the bit $\mathbb{1}\{\iota_3 \to \iota_1 \in \mathcal{E}_t\}$ in its residual stream; and (ii) Lem. B.1, which lets a padding symbol compute and look up layer-norm hashes of structured keys at constant depth.

We identify each intermediary $h_i(\iota)$ by the pair $(\iota, i)$ consisting of its parent item and its index in the chain, with $1 \leq i \leq N_{\max}$ for $N_{\max} \stackrel{\text{def}}{=} \mathcal{O}(N)$ the maximum possible fan-out (the number of edge-paddings allocated per item in block (1)). By Lem. B.1 applied to the schema $\mathcal{A} = \mathcal{I} \times \{1, \ldots, N_{\max}\}$, a constant-depth AHAT block writes, at the $(\iota, i)$-th intermediary padding symbol, the layer-norm hashes $\phi(\iota), \phi(i)$ into its residual stream. There are $|\mathcal{I}| \cdot N_{\max} = \mathcal{O}(N^2) \cdot \mathcal{O}(N) = \mathcal{O}(N^3)$ such padding symbols. Positions $(\iota, i)$ with $i > \text{out-deg}_{\mathcal{E}_t}(\iota) - 2$ (i.e., corresponding to possible neighbors that $\iota$ does *not* have) are ignored at iteration $t$: their residual stream carries no edges and they play no role in the pebble game.

To populate $\iota.\texttt{arg}_{\text{L}}, \iota.\texttt{arg}_{\text{R}}$ at every internal node of the binarized graph $\mathcal{T}(\mathcal{I}, \mathcal{E}_t)$, we first need to enumerate the active out-neighbors of each item $\iota$ in $\mathcal{E}_t$. Recall from block (1) that each candidate out-edge $\iota \to \iota'$ corresponds to a unique edge-padding symbol that, after block (1), stores a bit indicating whether the edge is currently active. A constant-depth attention block, using the $(\iota, i)$-indexing of Lem. B.1, computes prefix sums of these active-edge bits along the out-edges of each $\iota$ and assigns each active out-edge a dense rank $r \in \{1, \ldots, \text{out-deg}_{\mathcal{E}_t}(\iota)\}$; we write $\iota_r$ for the rank-$r$ active out-neighbor of $\iota$.

The pointers $\texttt{arg}_{\text{L}}, \texttt{arg}_{\text{R}}$ at each node of the binarized graph are then defined by the following table, populated in $\mathcal{O}(1)$ further layers by one attention head per pointer with an equality check on a $(\iota, \cdot)$ key:

$$
\begin{aligned}
&\iota \text{ with out-degree } M \geq 3: &&\texttt{arg}_{\text{L}} = \iota_1, &&\texttt{arg}_{\text{R}} = h_1(\iota), \\
&h_i(\iota) \text{ with } 1 \leq i \leq M - 3: &&\texttt{arg}_{\text{L}} = \iota_{i+1}, &&\texttt{arg}_{\text{R}} = h_{i+1}(\iota), \\
&h_{M-2}(\iota): &&\texttt{arg}_{\text{L}} = \iota_{M-1}, &&\texttt{arg}_{\text{R}} = \iota_M, \\
&\iota \text{ with out-degree } M = 2: &&\texttt{arg}_{\text{L}} = \iota_1, &&\texttt{arg}_{\text{R}} = \iota_2, \\
&\iota \text{ with out-degree } M = 1: &&\texttt{arg}_{\text{L}} = \iota_1, &&\texttt{arg}_{\text{R}} = \iota_1, \\
&\iota \text{ with out-degree } M = 0: &&\iota \text{ is a leaf, no pointers needed.}
\end{aligned}
$$

Concretely, each row is realized by one attention head whose query is the residual stream of the node on the left and whose

key matches the $(\iota, \cdot)$-encoding (via Lem. B.1) of the target on the right. The duplicate $\texttt{arg}_\texttt{R} = \iota_1$ in the $M = 1$ row evaluates $\vee$ with one operand twice, which is idempotent.

Each row of the table is filled in $\mathcal{O}(1)$ transformer layers, so the entire binarization block runs in $\mathcal{O}(1)$ depth, as required.

**Solving reachability queries.** This block implements steps (3)–(4) of the roadmap. The goal is to evaluate the unrolled marking formula Eq. (6) at every item $\iota \in \mathcal{I}$ in parallel and write the result $\mathbb{1}\{\iota \in \mathcal{I}_t\}$ into $\iota$'s item-padding slot, thereby implementing Eq. (4). Because the given grammar is unambiguous, for each item $\iota$, the reachable subgraph $\texttt{R}_t(\iota)$ from $\iota$ in $(\mathcal{I}, \mathcal{E}_t)$ (cf. Eq. (3)–Eq. (4)) becomes an arborescence rooted at $\iota$ (Fact 4.1). Reachability queries over binary trees now reduce to evaluating the Boolean formula associated with the binary tree. To invoke Lem. B.3 we first populate the residual stream of every node of the binarized dependency graph with the slots that the lemma's hypothesis requires. Original items $\iota$ that are currently realizable (i.e., $\iota \in \mathcal{I}_{t-1}$) and that appear as *leaves* of some $\texttt{R}_t(\iota)$ receive $\texttt{value} \overset{\text{def}}{=} \text{T}$; original items that appear as leaves of some $\texttt{R}_t(\iota)$ but are not currently realizable receive $\texttt{value} \overset{\text{def}}{=} \text{F}$. Every *internal* node of the binarized graph—both original items that have at least one out-neighbor in $(\mathcal{I}, \mathcal{E}_t)$ and the intermediary padding symbols introduced by $\mathcal{T}$—gets $\iota.\texttt{type} \overset{\text{def}}{=} \vee$ together with pointers $\iota.\texttt{arg}_\texttt{L}, \iota.\texttt{arg}_\texttt{R}$ to its two children in the binarized tree. These pointers are exactly the address lookups computed in the binarization step just above (and via Lem. B.1 for the original items at the root of each tree).

Lem. B.3, as stated, evaluates one expression tree rooted at one node, but its proof (§B.4) in fact runs the parallel pebble game over every internal node of the input graph simultaneously: at each iteration, the activate, square, and pebble operations are applied in parallel at all internal nodes, regardless of which root each node ultimately serves. We exploit this directly: all items' binarized reachable subgraphs $\{\texttt{R}_t(\iota)\}_{\iota \in \mathcal{I}}$ share the same set of $\mathcal{O}(N^3)$ padding symbols, and each item $\iota$ is simultaneously the root of its own tree $\texttt{R}_t(\iota)$ and an internal node or leaf of the trees rooted at the items that can reach it. A single pebble-game pass on this global graph therefore pebbles up $\iota.\texttt{value}$ at every root $\iota$ in $\mathcal{O}(\log N)$ steps in parallel, and we read off $\mathbb{1}\{\iota \in \mathcal{I}_t\}$ from $\iota.\texttt{value}$ at $\iota$'s own item-padding slot.

The above construction is consistent because the typing of each node is a property of the *graph* $\mathcal{T}(\mathcal{I}, \mathcal{E}_t)$, not a per-root property of the trees $\{\texttt{R}_t(\iota)\}_{\iota \in \mathcal{I}}$. The items in $\mathcal{I}_{t-1}$ are precisely the original items that are graph-leaves in $\mathcal{T}(\mathcal{I}, \mathcal{E}_t)$ and start the pass with $\texttt{value} = \text{T}$; the items outside $\mathcal{I}_{t-1}$ with no outgoing edge in $\mathcal{E}_t$ are graph-leaves with $\texttt{value} = \text{F}$; and every other node—original items with at least one out-edge in $\mathcal{E}_t$, and all intermediary padding symbols introduced by $\mathcal{T}$—is graph-internal, with $\texttt{value} = \bot$ and $\texttt{type} = \vee$ at the start of the pass. This typing matches the hypothesis of Lem. B.3 unambiguously, so a single pebble-game pass on $\mathcal{T}(\mathcal{I}, \mathcal{E}_t)$ fills $\texttt{value}$ at every graph-internal node $\iota$ in parallel with the value of the formula rooted at $\iota$, namely $\bigvee_{\iota' \in \texttt{R}_t(\iota)} \mathbb{1}\{\iota' \in \mathcal{I}_{t-1}\} = \mathbb{1}\{\iota \in \mathcal{I}_t\}$ by Eq. (6).

**Recognition step.** This block runs after the $\mathcal{O}(\log N)$ outer iterations of blocks (1)–(4) have completed. By Chytil et al. (1991), the marked set has then reached its fixpoint, $\mathcal{I}_t = \mathcal{I}_*$, and $\boldsymbol{w} \in L(\mathcal{G}) \iff (0, \text{S}, N] \in \mathcal{I}_* \iff$ the item-padding for $(0, \text{S}, N]$ stores T. The EOS symbol can attend to the item-padding associated with $(0, \text{S}, N]$ and check whether it is realizable, i.e., stores T in its residual stream.

**Depth and padding accounting.** Summing the costs of the four per-iteration blocks and the surrounding initial-items and recognition blocks gives the class $\text{MAHAT}_3^2$ claimed by the theorem:

- *Padding:* $\mathcal{O}(N^2)$ for item symbols, $\mathcal{O}(N^3)$ for right-witness edge symbols, $\mathcal{O}(N^3)$ for left-witness edge symbols, and $\mathcal{O}(N^3)$ for binarization intermediaries, summing to $\mathcal{O}(N^3)$.
- *Depth per outer iteration* (one pass through blocks (1)–(4)): $\mathcal{O}(1)$ for edge construction (Eq. (3)) + $\mathcal{O}(1)$ for binarization ($\mathcal{T}$) + $\mathcal{O}(\log N)$ for the pebble game of Lem. B.3 + $\mathcal{O}(1)$ for writing the bits of Eq. (4) = $\mathcal{O}(\log N)$.
- *Outer iterations:* $\mathcal{O}(\log N)$ by Chytil et al. (1991), after which $\mathcal{I}_t = \mathcal{I}_*$.
- *Total depth:* $\mathcal{O}(\log N) \cdot \mathcal{O}(\log N) = \mathcal{O}(\log^2(N))$, with constant-depth initial-items and recognition blocks contributing only an additive $\mathcal{O}(1)$.
- *Heads:* causally-masked for padding-to-string attention (positional encodings, layer-norm hash of Lem. B.1), unmasked for inter-padding-symbol attention (equality checks during edge construction, $\texttt{arg}_\texttt{L}, \texttt{arg}_\texttt{R}$ lookups in the pebble game, accept head). This places the construction in $\text{MAHAT}_3^2$. Applying Lem. 2.1 yields $\text{MAHAT}_3^2 \subseteq \text{AHAT}_4^2$, matching the theorem statement.

$\blacksquare$

**Theorem 4.2.** $\text{ULCFL} \subseteq \text{MAHAT}_2^1 \subseteq \text{AHAT}_3^1$.

*Proof.* Fix the input $w$ and an iteration index $t$. By linearity, the only rules that combine a non-terminal with the marked set $\mathcal{I}_{t-1}$ via $\mathcal{R}$ are of the form $A \to aB$ or $A \to Ba$, and the terminal sibling is necessarily of length 1. Hence the edges of $(\mathcal{I}, \mathcal{E}_t)$ take one of two shapes: an edge $(i, A, j] \to (i+1, B, j]$ whenever $A \to w_{i+1}B \in \mathcal{P}$ (using the witness $(i, A', i+1] \in \mathcal{I}_0$ with $A' \to w_{i+1} \in \mathcal{P}$), or symmetrically an edge $(i, A, j] \to (i, B, j-1]$ whenever $A \to Ba \in \mathcal{P}$. Each item therefore has at most $2|\mathcal{N}| = \mathcal{O}(1)$ outgoing edges in $\mathcal{E}_t$, one per applicable production. Since $|\mathcal{I}| = \mathcal{O}(N^2)$, and the $\mathcal{O}(1)$ out-degree gives $|\mathcal{E}_t| = \mathcal{O}(N^2)$, allocating one padding symbol per item and one per edge results in $\mathcal{O}(N^2)$ padding symbols.

**One reachability pass suffices.** We claim that $\mathcal{I}_1 = \mathcal{I}_*$, so that the outer loop of §4.1 collapses to a single iteration. $\mathcal{I}_1 \subseteq \mathcal{I}_*$ holds by definition. For $\mathcal{I}_* \subseteq \mathcal{I}_1$, suppose some item $\iota_3 = (i, A, j] \in \mathcal{I}_*$ is added at outer iteration $t \geq 1$ by an application of $\mathcal{R}$. By linearity, the rule responsible is either $A \to aB$ or $A \to Ba$. In the first case, $a = w_{i+1}$ and the witness is $(i, A', i+1] \in \mathcal{I}_0 \subseteq \mathcal{I}_{t-1}$ for some $A'$ with $A' \to w_{i+1} \in \mathcal{P}$, while the non-terminal child is $(i+1, B, j]$; the symmetric case is analogous with $a = w_j$ and witness $(j-1, A', j] \in \mathcal{I}_0$. Either way, the witness lies in $\mathcal{I}_0$, so the same combination triggers an edge from $\iota_3$ to $(i+1, B, j]$ (respectively $(i, B, j-1]$) *already in* $\mathcal{E}_1$ by Eq. (3). Conversely, every edge in $\mathcal{E}_1$ corresponds to a CKY combination by construction of $\mathcal{R}$, so reachability in $(\mathcal{I}, \mathcal{E}_1)$ is sound. Iterating this along any derivation of $\iota_3$ shows that $\iota_3$ reaches $\mathcal{I}_0$ in $(\mathcal{I}, \mathcal{E}_1)$, hence $\iota_3 \in \mathcal{I}_1$ by Eq. (4).

**Transformer construction.** The MAHAT construction of Thm. 4.1 applies directly; we can reuse its initial items, edge-construction, binarization, and pebble-game blocks. However, the outer loop collapses from $\log N$ iterations to a single one. This results in $\mathcal{O}(N^2)$ padding and $\mathcal{O}(\log N)$ looping, and applying Lem. 2.1 yields $\text{MAHAT}_2^1 \subseteq \text{AHAT}_{1+\max(2,1)}^1 = \text{AHAT}_3^1$. ∎

## C. Experimental Setup

**Data.** We used Anonymous's (2026) *length-constrained* sampling algorithm for CFLs to generate datasets. To sample strings from a given grammar $\mathcal{G}$, we consider the *probabilistic* version of $\mathcal{G}$ which induces a probability distribution over $L(\mathcal{G})$, which enables length-constrained sampling. Importantly, their procedure first samples a desired string length $N$, and then performs sampling over the distribution of all strings of length $N$. Negative strings are either sampled at random from $\Sigma^*$ or are perturbations from positive strings by applying random edits to them, the number of which is randomly sampled from a geometric distribution that favors small values (Butoi et al., 2025).

We therefore follow this procedure to sample positive and negative strings from handpicked context-free grammars except for BFVP. For BFVP, the negative strings are Boolean formulas that evaluate to F, enabling us to examine whether a transformer can correctly evaluate formulas rather than checking if an input is well-formed. We argue that the ability to process hierarchically nested structures such as nested subformulas in BFVP is already captured by the grammar $\text{Dyck}(k)$.

The training set consists of 1 million samples with string length at most 50. The test set has 2000 samples with string lengths at most 500. Testing the model on strings longer than those seen in training enabled the evaluation of its ability to *generalize* out-of-distribution.

**Models and Training Procedure.** We trained causally masked looped transformers with no positional embeddings. We used the PYTORCH implementation of a transformer encoder layer with pre-norm. Following our definition of the transformer in §2.2, we instantiated our models with an initial block of 2 transformer layers, a looping block (which is repeated $\log(N)$ times or once at inference) of 2 transformer layers and a final block of 2 transformer layers. A binary classifier (2 layer feedforward network) was then applied to the final contextual representation of EOS. Our transformers have 1.2 million parameter budget. We used the ADAMW optimizer (Loshchilov & Hutter, 2019) and binary cross-entropy loss, considering runs across 5 different seeds. The batch size was set to 64 and the learning rate to 0.0001.

