# OpenReview forum: "Context-free Recognition with Transformers"
_ICML.cc/2026/Conference — ICML 2026 regular_

### Official Review · Reviewer_aoTH · 2026-03-11

**Soundness:** 3
**Presentation:** 3
**Significance:** 3
**Originality:** 3
**Overall Recommendation:** 4
**Confidence:** 3

**Summary:**

This paper derives what formal requirements for looping transformers must have to be able to
accept context-free languages.
The paper shows that looping transformers with $O(n^6)$ padding tokens and $O(\log n)$ looping layers
are sufficient to verify whether a string $s$ of length $n$ belongs in a given CFL (context-free language)  $L$.
The paper constructs a parallel parsing algorithm to show this fact, that divides the question of whether
$s \in L$ into multiple sub-problems, where each sub-problem is solved in parallel.
The $O(n^6)$ padding tokens are required because this parallel algorithm can generate $O(n^4)$ `guesses', each of which require $O(n^2)$ space to solve, while the $O(\log n)$ looping layers
are needed because the membership query can be decomposed as a tree of subproblems of height $\log n$.
The paper then shows how the padding requirement can be reduced to $O(n^3)$ for unambiguous CFLs, taking
advantage of the fact that unambiguous CFLs do not require as many guesses when decomposing the membership
query into subproblems.
The paper then presents some experiments on whether looping transformers are efficient at recognizing CFLs.

**Compliance With Llm Reviewing Policy:**

Affirmed.

**Ethical Review Concerns:**

Thank you for the detailed response, which clarified most of my major concerns. I understand that the paper focuses on complexity from an asymptotic perspective, and agree with the authors that this paper makes a viable contribution in that direction. However, I am unfamiliar with recent developments in complexity theory, especially those for transformers, and thus cannot judge its overall relevance and impact to the overall field. My score thus remains a weak accept: I found the paper interesting and think it should be accepted if more expert reviewers confirm that the results make a relevant contribution to the field.

**Final Justification:**

I understand that the paper focuses on complexity from an asymptotic perspective, and agree with the authors that this paper makes a viable contribution in that direction. However, I am unfamiliar with recent developments in complexity theory, especially those for transformers, and thus cannot judge its overall relevance and impact to the overall field. I'm increasing my score to a weak accept: I found the paper interesting and think it should be accepted if more expert reviewers confirm that the results make a relevant contribution to the field.

**Key Questions For Authors:**

Do you know what the complexity bounds will look like when accounting for the CNF conversion?

It looks like the benchmarks are too easy (1 nonterminal?), could you consider an evaluation that illustrates the actual complexity in the size of the grammar?

**Limitations:**

yes, except those stated in my review

**Strengths And Weaknesses:**

Strengths:
- Interesting parallel algorithm to derive complexity bounds for transformers recognizing a CFL.
- The overall presentation is accessible: I did not have much trouble understanding how the overall bounds
  were derived, even without reading the proofs in detail.

Weaknesses:
- One of my main concerns with the paper is that it does not consider the blow-up of converting a CFL in general
  into CNF (Chomsky Normal Form), which I think can result in a $N^2$ increase where $N$ is the number
  of nonterminals in the grammar.
  The results of the paper are sound theoretically, but it remains questionable whether it is the string
  length $n$ or the size of the grammar $N$ that creates a bigger impact in practice.
  An experimental analysis including the grammar size would have made the results more compelling.
- I found the construction of the algorithm easy to follow at a high level, but was confused about
  how the algorithm can be captured in an actual transformer.
  The paper provides a proof sketch of how the algorithm can be embedded, but the short background
  in Section 2.2 and Appendix 1 did not provide with enough background to understand the embedding.
- It is unclear how the evaluation relates to rest of the paper; whether actual looping transformers are
  effective at recognizing CFLs seems to be an orthogonal question to the theoretical results of this
  paper.

---

> ### Author Rebuttal · Authors · 2026-03-31
>
> Thank you for reading our work and writing a constructive review!
> We address your feedback below.
>
> > it does not consider the blow-up of converting a CFL in general into CNF
>
> Our paper focuses on the impact of input length, which is how resources are measured in classical models of computation.
> However, Lemma 3.1. and Lemma 3.2 do state the space complexity in terms of |N|. E.g., for general CFL recognition, the space complexity is |N|^3 n^6. With an N^2 increase, we would have |N|^6 n^6 space required. We will revise the paper accordingly.
>
> > is the string length n or the size of the grammar N that creates a bigger impact in practice
>
> We believe this topic is out of scope as we focus on complexity from an asymptotic perspective. We would like to however reference some related works.
> In concurrent work to soon be released, increasing |N| decreases accuracy by a couple of hundredths. The CFLs are also randomly sampled, so we cannot infer the structural properties of the grammars.
> [1] finds that pre-trained LMs struggle on in-context CFL recognition when the number of production rules grows. Because the models are pre-trained, it is hard to make any conclusive theoretical conclusions.
>
> > It looks like the benchmarks are too easy (1 nonterminal?)
>
> We do not consider |N| as a measure of difficulty. Both Palindrome and Dyck1 do not require many non-terminals. However, Palindrome is inexpressible in C-RASP ([2]), but Dyck1 is provably ([2]) and empirically learnable.
>
> > confused about how the algorithm can be captured in an actual transformer
>
> We will clarify how our algorithm can be properly embedded in transformers.
> The proof is done by induction on items, where we show we can encode at position i the realizability of the associated item.
> A fresh dimension is reserved to indicate whether the associated item to a position is realizable.
>
> 1) At each position i, we can store $\phi(i)$. Recall we also allocate sufficiently many padding tokens for each 1) possible item and associated decomposition 2) each base case item. E.g., ([X,i,j],[Y,k,l]) can be associated with a padding token, where we will decompose [X,i,j] into [X,i,j]/[Y,k,l].
> Via Lemma B.1, we show how to store in the residual stream of each padding token their associated item and decomposition (cf. rebuttal with Reviewer icsp).
>
> 2) Base case:
>
> An [X,i,j] is a base case item iff i == j, and [X,i,j] is realizable iff X derives the symbol at position i. We show how to embed this on AHATs.
>
> $\phi(i)$ and $\phi(j)$ are both encoded in the residual stream. We can then perform an equality check for $\phi(i)$ and $\phi(j)$ using a feedforward network and ReLU activations.
> To then check if X can derive in one-step the symbol at position $i$, we can attend to the symbol at that position and copy its encoding.
> Finally, once we have stored in the padding token the symbol at $i$, a FFN can check if X (which is stored in the residual stream) can derive that symbol. This query is a mapping between two finite sets (non-terminals to symbols), and can therefore be computed by a FFN.
> The procedure to check if a slashed item [X,i,j] /[Y,k,l] is a realizable base case can be done using the same gadgets.
>
> 3) Induction step:
>
> A padding token stores 1) an item to solve (e.g., [X,i,j]) and 2) a set of objects that enable us to decompose that item (for instance, [Y,k,l]) (Lemma B.1).
> The resulting decomposition of [X,i,j]  is then the conjunction of [Y,k,l] and [X,i,j] /[Y,k,l].
> Via two attention layers, we can uniformly attend to all padding tokens associated with [Y,k,l] and [X,i,j]/[Y,k,l] and check if they are realizable.
> Importantly, if both [Y,k,l] and [X,i,j] /[Y,k,l] are realisable then [X,i,j] is as well. A FFN can compute the conjunction of two values.
>
> 4) The input string is in the language iff the item [S,1,n] is realizable.
> A FFN can add $\phi(1), \phi(S)$, and we also know how to add $\phi(n)$.
> We can then perform an equality check with all padding tokens that store $\phi(S), \phi(1)$ and $\phi(n)$.
> It suffices for only one of these padding tokens to store the value for realizability.
>
> Please let us know if any concerns on your behalf remain.
>
> > Whether actual looping transformers are effective at recognizing CFLs seems to be an orthogonal question
>
> Our theory makes testable claims about the ability of various transformers to recognize CFLs with looping and padding; the theory sets out to test whether some of these constructions can actually be learned in practice.
> E.g., we make the testable claim that looping helps recognize BFVP.
> Foundational work ([3]) demonstrates general CFL recognition requires log-depth circuits. Analogously, we show log-depth transformers can recognize CFLs. Thus, validating whether looping helps process syntax in practice vets our theory.
>
> We also invite you to read our response to Reviewer icsp where we show linear padding empirically improves performance.
>
> [1] Petty et al 2025
> [2] Yang et al 2024
> [3] Venkateswaran 1991

---

> > ### Author Rebuttal · Reviewer_aoTH · 2026-04-03
> >
> > Thank you for the detailed response, which clarified most of my major concerns. I understand that the paper focuses on complexity from an asymptotic perspective, and agree with the authors that this paper makes a viable contribution in that direction. However, I am unfamiliar with recent developments in complexity theory, especially those for transformers, and thus cannot judge its overall relevance and impact to the overall field. I'm increasing my score to a weak accept: I found the paper interesting and think it should be accepted if more expert reviewers confirm that the results make a relevant contribution to the field.

---

### Official Review · Reviewer_eDHf · 2026-03-13

**Soundness:** 3
**Presentation:** 3
**Significance:** 2
**Originality:** 3
**Overall Recommendation:** 4
**Confidence:** 2

**Summary:**

Authors show that looped- and padded-transformers with O(log(n)) looping layers and O(n^6) padding tokens can recognise all CFL by a rigorous construction and utilising parallel parsing.  Also for natural subclasses such as unambiguous CFLs, authors show that the recognition problem on transformers requires only O(n^3) padding.

**Compliance With Llm Reviewing Policy:**

Affirmed.

**Final Justification:**

This paper provides interesting theoretical insights into transformers by rigorous construction of looped transformers that can implement parallel parsing.

**Key Questions For Authors:**

line 30: "However, such serial procedures cannot be naturally implemented due to transformers’ highly parallel, fixed-depth structure"
Why highly parallel structure prevents implementing serial procedure?

Would standard transformer with CoT be able to recognise CFLs?

**Limitations:**

yes

**Strengths And Weaknesses:**

This work provides interesting theoretical insights into transformers by rigorous construction of looped transformers that can implement parallel parsing. It provides algorithm, prove correctness and bounds.
This work complements recent interpretability results.

Even it is theoretically possible to construct such transformer, It is not clear if possible to learn in in practice.
Furthermore, while experimental results show some improvements with log(n) looping, the performance in OOD settings is not sufficiently convincing to suggest that looping is a fundamental requirement.

---

> ### Author Rebuttal · Authors · 2026-03-31
>
> Thank you for taking the time to read our work and writing a constructive review.
> We address your feedback below.
>
> > Even it is theoretically possible to construct such transformer, It is not clear if possible to learn in practice.
>
> In practice, fixed-depth transformers learn a similar parallel algorithm to our theoretical construction when parsing strings by implicitly encoding items of the form [X,i,j] ([1]), which leads us to believe that our construction can be learnt in practice.
>
> However, when given samples from a specific language, a transformer usually learns a more fine-grained program rather than a general CFL parsing procedure like ours ([10]). We still believe our work advances our understanding of the ability of LMs to process syntax by understanding which minimal extension allows for processing CFL classes.
>
> > Furthermore, while experimental results show some improvements with log(n) looping, the performance in OOD settings is not sufficiently convincing to suggest that looping is a fundamental requirement.
>
> We ran extra experiments (cf. rebuttal with Reviewer icsp) with longer OOD strings to address your remark, and found that over longer strings, looping and padding increases accuracy by around 5 hundredths, which is in line with our theory.
>
> We do not necessarily claim that looping will improve performance over any CFL. For instance, for languages in C-RASP—which are provably learnable and expressible by fixed-depth transformers, we conjecture looping might not help by much.
> On the other hand, we make the testable claim that looping should help solve BFVP— a statement we empirically vet.
>
> > Why highly parallel structure prevents implementing serial procedure?
>
> As formalized in circuit complexity, some computational problems are believed to be inherently sequential, meaning they cannot be implemented by parallel computational models like transformers ([3]).
>
> The circuit-complexity class TC0 is the class of problems that are efficiently solvable in parallel in constant time. Fixed-depth transformers are known to sit in this class ([3]).
> On the other hand, we consider an algorithm to be serial when it requires non-constant time.
> Problems that require non-constant time sit beyond TC0 (e.g., regular language recognition is in NC1). CFL recognition sits between NC1 and AC1 ([4]).
> Therefore, while fixed-depth transformers’ inherent parallel structure enables them to efficiently solve problems in TC0, they cannot solve inherently serial problems beyond TC0.
> As CFL recognition is one of these problems, we consider an extension of transformers with dynamically-scaling time and space ([5]). As we prove, log(n) layers enables expressing problems beyond TC0 at minimal sequential cost.
>
> > Would standard transformer with CoT be able to recognise CFLs?
> Transformers with CoT can recognize CFLs, though it would likely require a polynomial amount of CoT in contrast to a minimal (logarithmic/sublinear) amount of looping.
>
> Both softmax and AHAT transformers with CoT are Turing-complete ([6], [7]) and can therefore do general CFL recognition assuming we can allocate a number of decoding steps polynomial in input length.
> However, if we bound the amount of decoding steps (w.r.t. input length n) in CoT to a more realistic setting, we get tighter bounds on what CoT transformers can achieve. O(log(n)) CoT steps do not increase expressive power much beyond TC0 ([6]), and therefore probably cannot solve general CFL recognition under standard complexity conjectures. In contrast, we are showing that with O(log n) looping steps, transformers can recognize CFLs (with padding tokens as well, which are fully parallelizable).
>
> Finally, we believe looped-transformers can solve inherently parallel problems more efficient than CoT. It has been shown that for regular language recognition, looping is significantly more efficient than scaling CoT steps ([8]). We argue that the same phenomenon should hold for CFLs. CoT is inherently more sequential in nature than looping as it samples a token step-by-step. Meanwhile, looping dynamically increases the number of times we update the contextual representations, which exploits transformers’ inherent parallelism.
> Moreover, [9] finds CoT transformers struggle on in-context CFL recognition, requiring up to O(n^6) decoding steps—a bound similar to our O(n^6) padding tokens.
>
> [1] Zhu et al 2023
> [2] Huang et al 2024
> [3] Merrill et al 2022
> [4] Venkateswaran 1991
> [5] Merrill et al 2024
> [6] Merrill et al 2024
> [7] Jiang et al 2025
> [8] Merrill et al 2025
> [9] Petty et al 2025
> [10] Huang et al 2026

---

> > ### Author Rebuttal · Reviewer_eDHf · 2026-04-04
> >
> > Thank you for the detailed response.

---

### Official Review · Reviewer_efr7 · 2026-03-18

**Soundness:** 3
**Presentation:** 4
**Significance:** 3
**Originality:** 3
**Overall Recommendation:** 5
**Confidence:** 2

**Summary:**

This paper focuses on the problem of using transformers to recognize context-free languages (CFLs), which is impossible for fixed-size transformers under standard complexity conjectures. Previous work showed that the addition of $O(\log n)$ looping layers enable transformers to recognize regular languages, where n is the input length. However, standard algorithms for recognizing CFLs such as CKY are sequential and would be expected to require exponentially more looping layers. The authors demonstrate theoretically that $O(\log n)$ looping layers, along with $O(n^6)$ padding tokens, are enough for transformers to implement parallel CFL recognition algorithms and thereby recognize all CFLs. This reduces to $O(n^3)$ padding tokens for *unambiguous* CFLs (meaning only one possible parse exists), and further down to $O(n^2)$ for *linear* unambiguous CFLs (where each grammar rule has at most one open subproblem). These theoretical results are validated empirically on variable-free Boolean formula value problems (BFVP), which are known to require logarithmic depth, along with other CFLs such as palindromes, marked palindromes, and Dyck languages.

**Compliance With Llm Reviewing Policy:**

Affirmed.

**Final Justification:**

The rebuttal addressed my concerns about the paper's evaluation, so I have increased my score. I will also flag that my confidence is at 2 since my ability to assess the correctness of the proofs is limited.

**Key Questions For Authors:**

1. It is my understanding that the complexity conjectures on the ability for fixed-size transformers to recognize CFLs are asymptotic claims. Is it possible that the performance gap between the fixed and looped transformers would be wider if the input length was increased? If so, a demonstration of this would be informative.
2. Is there a setting (e.g., a particular grammar, or longer input lengths) where adding padding tokens measurably improves trained model performance?
3. Since the BFVP experiments are intended to confirm the hypothesis that $O(\log n)$ looping layers are needed, would it be possible to include a sweep over intermediate loop iteration counts? A gradual degradation in performance as loops are removed would be more informative than the current comparison (fixed-size vs. log n).

**Limitations:**

Yes

**Strengths And Weaknesses:**

Full disclosure, my expertise is not in theoretical computer science and I have calibrated my review accordingly.
### Strengths

1. The theoretical results are well presented as a tradeoff between the number of looping layers and the number of padding tokens, put in a hierarchy of CFL subclasses. This is more informative than just a yes/no answer on whether CFLs can be recognized or not.
2. The result that unambiguity dramatically reduces the number of padding tokens needed for recognition is a particularly elegant result, and motivated well alongside prior empirical work on transformers struggling to parse ambiguous grammars.
3. The paper is well written with conclusions given upfront, which is noteworthy given that many of the key contributions are more theoretically dense.

### Weaknesses

1. The number of padding tokens appears to be central to the theoretical results of the paper, but the experiments are run on CFLs that do not require extra padding for recognition. The role of looping is validated with the BFVP experiments, but the role of padding tokens seems absent from the evaluations.
2. Looping is shown to provide only slight improvements (1-3% in-distribution, 3-4% out-of-distribution) over not looping at all. This is acknowledged in the paper, but it would be stronger if accompanied with hypotheses on how the performance is competitive (88%) without looping.

---

> ### Author Rebuttal · Authors · 2026-03-31
>
> Thank you for taking the time to read our work and writing a constructive review.
> We address your feedback below.
>
> > The number of padding tokens appears to be central to the theoretical results of the paper, but the experiments are run on CFLs that do not require extra padding for recognition. The role of looping is validated with the BFVP experiments, but the role of padding tokens seems absent from the evaluations.
>
> We ran some additional experiments to clarify your concerns.
> We consider OOD strings up to lengths 400. We consider the languages Dyck2 and Palindrome. We get the following results.
>
> Palindrome: fixed-depth 0.68, looping 0.67, looping+linear padding 0.72.
>
> Dyck2: fixed-depth 0.83, looping 0.84, looping+linear padding 0.87
>
> Interestingly, over longer OOD strings, while the gap between only looping and fixed-depth is quite small, both looping and linear padding offers better accuracy (by around 5 hundredths), empirically vetting the increase in performance padding can confer.
> We will revise the paper with such experiments.
>
> > Looping is shown to provide only slight improvements (1-3% in-distribution, 3-4% out-of-distribution) over not looping at all. This is acknowledged in the paper, but it would be stronger if accompanied with hypotheses on how the performance is competitive (88%) without looping.
>
> In the paper, we addressed for every language why a fixed-depth transformer might perform well on a given CFL. To this extent, the C-RASP conjecture—the class of languages transformers should express and learn successfully, offers a hypothesis for which CFLs transformers should perform well on. For instance, our fixed-depth transformers perform well on Dyck, which is supported by the fact that C-RASP allows for counting—a mechanism required to solve Dyck. On the other hand, Palindrome is not in C-RASP, which explains why our fixed-depth transformers struggle on that language.
>
> > It is my understanding that the complexity conjectures on the ability for fixed-size transformers to recognize CFLs are asymptotic claims. Is it possible that the performance gap between the fixed and looped transformers would be wider if the input length was increased? If so, a demonstration of this would be informative.
>
> We already tested length generalization in the paper. Referencing our previous response, on longer OOD strings, having both looping and linear padding offers a quantifiable gain in performance, while the gap in performance between just looping and fixed-depth is tightened. This still aligns with our theory, which argues both padding and looping is necessary.
>
> > Is there a setting (e.g., a particular grammar, or longer input lengths) where adding padding tokens measurably improves trained model performance?
>
> As we detailed previously, having both looping and padding improves performance over Dyck2 and Palindrome. For Palindrome, we conjecture that linear padding can help the model understand where the split point is in the string.
>
> > Since the BFVP experiments are intended to confirm the hypothesis that  looping layers are needed, would it be possible to include a sweep over intermediate loop iteration counts? A gradual degradation in performance as loops are removed would be more informative than the current comparison (fixed-size vs. log n).
>
> Our models are trained to leverage log(n) looping, and we naturally suspect that doing inference with, say log(n)-1 looping will result in decreasing performance and is uninformative. Our goal in the paper is to measure the gap in performance between fixed-depth and looped transformers.

---

> > ### Author Rebuttal · Reviewer_efr7 · 2026-04-03
> >
> > The additional padding experiments on Dyck2 and Palindrome directly address my primary evaluation concern, and the results appear consistent with the theoretical claims. The C-RASP explanation for how fixed-depth transformers remain competitive is also helpful.
> >
> > I am raising my score to reflect that these concerns have been resolved. For the camera-ready, I would suggest adding at least one more language in and out of C-RASP that requires padding tokens, so that the conclusions don't rest on a single example per category.

---

### Official Review · Reviewer_icsp · 2026-03-19

**Soundness:** 3
**Presentation:** 3
**Significance:** 2
**Originality:** 2
**Overall Recommendation:** 4
**Confidence:** 3

**Summary:**

This paper aims to theoretically show that transformers can recognize context-free languages (CFLs). The problem addressed is interesting, as syntax is traditionally modeled using CFLs, and a theoretical proof that transformers can recognize CFLs would provide a useful explanation for their empirical success.

However, the value of any theoretical result depends on the soundness of its modeling assumptions. In this work, several assumptions appear overly idealized. For example, the use of AHAT assumes exact max-selection and uniform averaging in the case of ties, which are critical for implementing precise branching behavior. The authors also assume log-precision arithmetic, which is difficult to realize in the inherently noisy floating-point environment of neural networks. Taken together, these assumptions make the model closer to a parallel algorithmic machine than to a practical neural architecture.

This work builds upon classical parallel algorithms for CFL recognition (Ruzzo 1980; Chytil et al. 1991) and adapts these ideas to the transformer setting. The main novelty lies in expressing these constructions within the transformer framework and analyzing the required resources (looping and padding), rather than introducing fundamentally new algorithms for CFL recognition.

Overall, I believe the assumptions are sufficiently idealized that the results should be interpreted primarily as theoretical expressivity statements rather than practical capabilities of real transformers. That said, given the importance of transformers in modern machine learning, adapting classical theoretical results to this setting is valuable and informative. In particular, the paper clarifies what assumptions are required for CFL recognition within the transformer framework.

**Compliance With Llm Reviewing Policy:**

Affirmed.

**Key Questions For Authors:**

- In line 676, $\phi(z)$ is defined as a real-valued vector normalized by a scalar factor. Later, in Lemma B.1 (line 702), $phi(i)$ is treated as an integer and used in modulus operations, which seems incompatible with the earlier definition and would render the proof incorrect. Can you explain this?

- You provide explicit upper bounds on the required looping and padding for several CFL classes. I'm curious about any tightness results, lower bounds, or evidence that these resource bounds are close to necessary in their model? Is $O(n^6)$ padding for general CFL recognition an artifact of the current construction, or is there reason to think something of this order is unavoidable?

**Limitations:**

yes

**Strengths And Weaknesses:**

Strengths

- The core question of: under what assumptions transformers can recognize CFLs and what trade-offs are involved, is relevant, interesting and insightful.

- Applying classical parallel algorithms for CFL recognition to the transformer setting is a meaningful contribution. In particular, the paper clarifies the roles of looping (as sequential computation) and padding (as memory), and provides explicit resource bounds for different CFL subclasses. These are valuable and concrete contributions.

- The paper is well-structured and clearly written, with a logical progression from general CFLs to restricted subclasses.

Weaknesses

- The modeling assumptions are highly idealized. In particular, the use of AHAT, log-precision arithmetic, and polynomially many padding tokens makes the model closer to a parallel algorithmic machine than to a practical neural architecture, which limits the applicability of the results to real-world transformers.

- While the paper does not address learnability, I do not consider this a major weakness, as the expressivity analysis is a good enough contribution.

- The experimental validation is relatively weak. The experiments do not directly correspond to the main theoretical constructions and provide limited insight into their practical implications. However, since this is a theoretical work in nature, I don't see this as a major limitation.

- Regarding the proofs, some parts of the appendix still read more like sketches than fully detailed constructions. In particular, Lemma B.1 was difficult to follow, and I was not able to fully verify its correctness. Correctness of this proof would be critical in my final recommendation.

---

> ### Author Rebuttal · Authors · 2026-03-31
>
> Thank you for taking the time to read our work and writing a constructive review.
> We address your feedback below.
>
> > The modeling assumptions are highly idealized
>
> **Padding**. We view padding as an extension for transformers that can improve performance in a variety of tasks, e.g. CFL recognition.
> Our goal is not to explain how fixed-depth transformers process syntax, but rather understand what minimal extension provides the resources to do so.
>
> **Log-precision**. An integer n requires log(n) bits to be represented. Therefore, because language models mostly manipulate representable values, log-precision reflects the amount of bits used in practice. This is vetted by the CRASP conjecture ([1]), verified experimentally ([2]).
>
> **AHAT**. We agree that AHAT does not literally correspond to transformers in practice, but believe it is a useful model to study realistic transformers. With fixed-precision, AHAT and softmax are equivalent ([3]). With log-precision, transformers can approximate AHAT up to a context length by learning a large temperature (w.r.t. n) ([12]). In practice, norm growth during training biases transformers to AHAT ([4]).
>
> > While the paper does not address learnability...
>
> We agree learnability is out of scope. Future work may benefit from our results: e.g., expressibility results for Parity led to results on the inability of transformers to learn it ([5],[6]).
>
> > The experimental validation is relatively weak…
>
> Our theory predicts looped-transformers outperform fixed-depth on BFVP—a prediction we empirically vetted.
> We acknowledge our paper lacks padding experiments and have ran some more experiments:
>
> We consider OOD strings up to lengths 400.
> We consider the languages Dyck2 and Palindrome.
>
> Palindrome: fixed-depth 0.68, looping 0.67, looping+linear padding 0.72
>
> Dyck2: fixed-depth 0.83, looping 0.84, looping+linear padding 0.87
>
> Padding does seem to improve performance, and we will make sure to add such experiments to our paper.
>
> > In particular, Lemma B.1 was difficult to follow…
>
> Let $S$ be a finite-set of integers representable with $log(n)$ bits.
> Importantly, $S$ can be then used to represent an item of the form $[X,i,j]$. ($X$, $i$ and $j$ can be represented with $log(n)$-bits)
>
> We denote by $L_S$ the number of distinct ways to write $S$ w.r.t. input length $n$. E.g., if $S$ corresponds to items $[X,i,j]$, then $L_S = |N| n^2$. By construction, we allocate $L_S$ padding tokens.
> We denote by $S_i$ the tuple associated with padding token $i$.
> Our lemma claims that for each of the $L_S$ padding tokens, we can store in the residual stream each element of its associated tuple $S_i$.
>
> The proof relies on [11]’s Lemma 1: with finitely many attention layers, given $\phi(i)$ and $\phi(n)$ we can store $\phi(i // n)$ and $\phi(i \mod n)$.
> $\phi(i)$ requires $\log(L_S)$ bits to be stored as there are $L_S$ padding tokens. We now show how to add each element of $S_i$ to the residual stream.
>
> E.g., assume the tuples $S_i$ contain two indices $p$, $q$ and a non-terminal $X$. The bits used to represent $i$ can then be partitioned into bits representing $p$, $q$ and $X$ (because $i$ is representable with $\log(L_S) = \log(|N|) + \log(n) + \log(n)$ bits). To store $p$, we can extract the first $\log(n)$ bits of $i$ by equivalently computing $i \mod n$ as we can pre-compute $\phi(i)$ and $\phi(n)$. We can now leverage [11] to add $\phi(i \mod n)$ to the residual stream at $i$. Similarly, to extract the following $\log(n)$ bits for $q$, we can clear out the initial $\log(n)$ bits by adding to the residual stream $\phi(i / n)$([11]).
>
> We will revise Lemma B.1 accordingly.
> We hope this explanation is more clear, let us know if anything else is unclear.
>
> > I'm curious about any tightness results…
>
> We first provide some insights on the bottleneck of our algorithm.
>
> A classic result in parsing is that CKY can be reduced to Boolean Matrix Multiplication (BMM).
> We have an analogous bottleneck with our algorithm—a parallel version of the parsing problem. We can re-write Algorithm 2 as:
> $\[X,i,j\] \textbackslash \[Y,k,l\] = \bigvee_{\[Z,p,q\]} \[X,i,j\] \textbackslash \[Z,p,q\] \land [Z,p,q\] \textbackslash \[Y,k,l\]$.
> This is essentially the formula for BMM between two Boolean matrices:
> $c_{ij} = \bigvee_{k=1}^m a_{ik} \land b_{kj}$
> While we still have not proved tightness, we find the parallel between CFL algorithms and matrix multiplication as a bottleneck interesting!
>
> We would also like to argue that padding is provably necessary. Looped- and padded-transformers are equivalent to AC-circuits ([9])—the machinery capable of processing CFLs. Padding corresponds to space complexity (circuit size) and is therefore required for CFL recognition.
>
> [1] Yang et al 2024
> [2] Huang et al 2024
> [3] Li et al 2025
> [4] Merrill et al 2021
> [5] Chiang et al 2022
> [6] Hahn et al 2024
> [7] Huang et al 2026
> [8] Valiant 1974
> [9] Merrill et al 2024
> [10] Rossmanith et al 1992
> [11] Merrill et al 2025
> [12] Yang et al 2025

---

> > ### Author Rebuttal · Reviewer_icsp · 2026-04-08
> >
> > Thank you for addressing my questions. I recommend acceptance.

---

### Decision · Program_Chairs · 2026-04-30

**Decision:**

Accept (regular)

**Comment:**

This is another expressivity paper with a very narrow scope and limited novelty regarding context-free language recognition. I recommend weak acceptance.